# Progesterone activation of β₁-containing BK channels involves two binding sites

Kelsey C. North ⬡ [1], Andrew A. Shaw ⬡ [1], Anna N. Bukiya[1] & Alex M. Dopico ⬡ [1] ✉

Progesterone (≥1 μM) is used in recovery of cerebral ischemia, an effect likely contributed to by cerebrovascular dilation. The targets of this progesterone action are unknown. We report that micromolar (μM) progesterone activates mouse cerebrovascular myocyte BK channels; this action is lost in $β_1^{-/-}$ mice myocytes and in lipid bilayers containing BK α subunit homomeric channels but sustained on $β_1/β_4$-containing heteromers. Progesterone binds to both regulatory subunits, involving two steroid binding sites conserved in $β_1$-$β_4$: high-affinity (sub-μM), which involves Trp87 in $β_1$ loop, and low-affinity (μM) defined by TM1 Tyr32 and TM2 Trp163. Thus progesterone, but not its oxime, bridges TM1-TM2. Mutation of the high-affinity site blunts channel activation by progesterone underscoring a permissive role of the high-affinity site: progesterone binding to this site enables steroid binding at the low-affinity site, which activates the channel. In support of our model, cerebrovascular dilation evoked by μM progesterone is lost by mutating Tyr32 or Trp163 in $β_1$ whereas these mutations do not affect alcohol-induced cerebrovascular constriction. Furthermore, this alcohol action is effectively counteracted both in vitro and in vivo by progesterone but not by its oxime.

Direct interaction between steroids and ion channel proteins, and its contribution to physiology, pathology and human therapeutics, constitute a growing area of research[1–3]. Large conductance, Ca²⁺- and voltage-gated K⁺ (BK) channels control essential physiological processes, including neuronal excitability, neurotransmitter release, circadian rhythms, the immunological response, and vascular smooth muscle (SM) tone[4–7]. BK channels result from the association of four channel-forming α (or slo1) subunits which show a ubiquitous tissue distribution[8]. In most mammalian tissues, however, these tetramers are associated with small, regulatory subunits: $β_{1-4}$, all having two transmembrane domains (TMs), and/or leucine rich repeat domain-containing proteins having one TM, with both regulatory families showing a rather selective cell-specific expression. Thus, BK regulatory subunits suit the resulting channel heteromers to tissue-specific physiology and constitute an appealing platform for developing drugs that target tissue-specific pathology driven by BK channel dysfunction[9–12].

Vascular SM BK channels, including those found in cerebral arteries, are highly enriched in regulatory $β_1$ subunits[13–15]. Remarkably, $β_1$ is required for BK channel activation by physiological or supra-physiological (therapeutic) concentrations of some steroids, such as bile acids and related cholanes that evoke vasodilation by their direct docking onto a site that is unique to $β_1$ TM2[16,17]. In turn, the 17β-estradiol site is different from the cholane site, and the key amino acids involved in estrogen-recognition are common to all βs[18]. In contrast, the pregnane steroid pregnenolone[19] and its precursor cholesterol[20] are able to modulate BK channel function independently of regulatory subunits.

Progesterone (PROG) is a pregnane-based hormone that, as its analog pregnenolone, can act as a local neurosteroid. In the brain, PROG plays key roles in neurogenesis, neuroregeneration, memory and cognition[21,22]. Remarkably, at supraphysiological concentrations (μM–mM) PROG is used in recovery from traumatic brain injury[23] and brain ischemic events[24], a therapeutic effect likely contributed to by cerebrovascular dilation. Given the key role of SM BK channels in evoking cerebrovascular dilation, and the fact that PROG was reported to activate heterologously expressed, recombinant $β_4$-containing BK

[1]Department of Pharmacology, Addiction Science, and Toxicology, College of Medicine, The University of Tennessee Health Science Center, Memphis, TN 38103, USA. ✉ e-mail: adopico@uthsc.edu

channels[25], we set to determine whether therapeutic levels of PROG activate cerebrovascular BK channels and, if so, pinpoint the specific site(s) of PROG action.

Combining microscale thermophoresis (MST), computational docking and molecular dynamics, engineered BK channel subunit constructs (chimeras; point mutants), the *KCNMB1*[-/-] (BK $\beta_1$ K/O) mouse model, reverse permeabilization of engineered subunit constructs in isolated arterial segments, patch-clamp methods on native channels, lipid bilayer electrophysiology on engineered channels, and evaluation of PROG action on brain arteries in vivo, we demonstrate that PROG at both naturally occurring and therapeutically relevant concentrations, binds sequentially to two sites in BK $\beta_1$. The site of lower affinity ($\mu$M range) wherein a single PROG molecule bridges the two TMs, is the effector of PROG-induced channel activation. This site is conserved throughout all $\beta$ subunits and thus, PROG also activates BK containing the neuronally-abundant $\beta_4$. In turn, the high-affinity PROG site plays a permissive role in facilitating PROG binding to the effector site but does not mediate steroid activation on its own. Consistent with our model, $\mu$M PROG but not a close structural analog that fails to dock onto the effector site, evokes cerebrovascular dilation and effectively blunts alcohol-induced cerebrovascular constriction, both in vitro and in vivo. Thus, we identified sites for steroid action on $\beta$-containing BK channels and their differential contribution to PROG action on channel function.

## Results

### Progesterone binds to and activates BK channels through their $\beta_1$ subunits

There are several reports of steroid hormones being able to directly modulate the activity of BK channels: 17$\beta$-estradiol and bile acids may activate BK channels through direct steroid-$\beta_1$ subunit interactions, albeit involving different steroid-recognition docking sites[17,18]. In contrast, pregnenolone, a pregnane steroid as PROG, modulates BK channel activity (in this case, inhibition) in absence of regulatory subunits[19,26]. PROG itself was reported to activate recombinant BK channels expressed in CHO cells when $\beta_4$ subunits are present[25]. Thus, we determined whether PROG molecules actually bound to the channel regulatory subunits, focusing on $\beta_1$, given its pathophysiological and potential therapeutical relevance as it is highly expressed in cerebrovascular SM[13–15]. We chose MST as this methodology is optimal for detecting and quantifying biomolecule interactions by the thermophoretic detection of time-dependent changes in protein intrinsic fluorescence related to conformation, charge, and size of a molecule as they are induced by a binding event[27]. Our MST data show clustered differences in temperature (delta, $\Delta$) detected between the start of thermal-induced conformational changes (onset) and peak change (inflection points) for the $\beta_1$ subunits that were subjected to a heat–cool cycle in an attempt to enhance conformational variability (boiled $\beta_1$ in Supplementary Fig. 1a, b). Additionally, no onset or inflection points are detected when no protein is present (Supplementary Fig. 1c), validating our MST assays. We tested a concentration range of PROG that spanned the hormonal fluctuations throughout the female cycle and during pregnancy, as well as following therapeutic administration of this steroid (0.004–40 $\mu$M) to construct a PROG concentration-binding curve to $\beta_1$ protein (Fig. 1a, b; reduced Chi-squared=0.076; 0.053, R-squared=0.99; 0.99 for 0.004-0.4 and 0.4-40 $\mu$M, respectively). Comparison of PROG to control vehicle (DMSO) shows that PROG decreases the difference in delta between the onset inflection events; this delta reflects PROG effect on thermally-driven conformational unfolding of the $\beta_1$ protein[28] (Fig. 1a). The plot of the $\Delta$ parameter (indicator of ligand binding; see Methods) as function of PROG concentrations shows a monotonic process with points that can be fit to a single exponential from 0.001 to 0.4 $\mu$M PROG. This "high-affinity" binding process reaches a maximum and a 50% value at 0.4-1 and 0.04 $\mu$M PROG, respectively. A second monotonic process with

points that can also be fit to a single exponential is evident at concentration of 0.4 $\mu$M PROG and higher. This "low-affinity" binding process reaches a maximum and a 50% value at 10 and ~2.2 $\mu$M PROG, respectively.

Following these biochemical data, we determined their functional impact by evaluating PROG action on the steady-state activity (NPo; see Methods) of slo1 (cbv1)+$\beta_1$ heteromeric channels reconstituted into a binary phospholipid planar bilayer (POPE/POPS 3:1 w/w), with *trans* and *cis* solutions containing 30 $\mu$M free [Ca$^{2+}$] (Fig. 1d–f). This system avoids the complex proteo-lipid environment that is characteristic of natural cell membranes and thus allows crude control of both channel protein and bilayer lipid components in the system to address the functional impact of ligand-ion channel protein *direct* interactions. PROG activated these heteromeric channels in a concentration-dependent manner (Fig. 1e–f; reduced Chi-squared=3.02, *R*-squared=0.98, exponential fit), with an EC$_{max}$ = 10 $\mu$M. However, this concentration response curve (CRC) differed from the binding data: PROG activated the channel only at concentrations greater than 0.1 $\mu$M, indicating that the low-affinity binding site, but not the high-affinity binding site, likely mediates PROG action on channel activity.

Additionally, to determine any possible contribution of BK $\alpha$ subunits (cbv1 isoform) to channel activation by high concentrations of PROG, we evaluated the effect of 10 $\mu$M PROG on cbv1 homo-tetramers expressed in lipid bilayers under conditions identical to those used to probe cbv1 + $\beta_1$ channels with PROG. Even 10 min after application, 10 $\mu$M PROG consistently failed to activate the homomeric channels, an output identical to that of applying DMSO-containing solution (control) (Fig. 1g–i). The differential effects of PROG and DMSO on BK $\alpha \pm \beta_1$ channels are summarized in Fig. 1l.

### Progesterone docks onto protein regions conserved between $\beta_1$ and $\beta_4$ to activate BK channels

Next, to begin to identify the functional binding site(s) for PROG in $\beta_1$ subunits, we compared the effect of PROG at its maximally effective concentration (10 $\mu$M) on the activity of cbv1 + $\beta_1$ vs. cbv1 + $\beta_4$ in parallel bilayer reconstitution experiments under identical recording conditions, as described the previous section. We chose $\beta_1$ and $\beta_4$ for the comparison because these subunits are the least conserved in primary sequence among all known $\beta$ subunits[13,14,29]: similar activation of these subunits by PROG would lead us to focus on a small number of residues that are conserved among the two as possible PROG recognition sites. In contrast to other steroids, such as cholanes, which only activate $\beta_1$-containing BK channels[16], PROG readily and robustly activated both slo1 + $\beta_1$ and slo1 + $\beta_4$ channels reconstituted into binary lipid bilayers (Fig. 1j–l; *P* = 0.0419 and 0.0449 for $\beta_1$ and $\beta_4$, respectively). Moreover, the PROG concentrations optimal to bind $\beta_1$ as determined by MST (4–10 $\mu$M; Fig. 1b, c) exerted a similar effect on the thermal unfolding of $\beta_4$ (Fig. 1c), raising the hypothesis that PROG binding involves residue(s)/site(s) conserved between these two regulatory subunits.

Following the biochemical data proving that $\mu$M PROG binds to isolated $\beta_1$ or $\beta_4$ subunits, and directly activates cbv1 + $\beta_1$/$\beta_4$ heteromeric channels, we used computational modeling on $\beta_1$ (*KCNMB1* product; NP_011452) and $\beta_4$ (*KCNMB4*; product; NP_055320) proteins (Supplementary Fig. 2) to dock PROG onto all atoms of their two TMs since (a) PROG is highly hydrophobic, and (b) both 17$\beta$-estradiol[18] and cholanes[16,17], which require $\beta_1$ to activate BK channels, were identified to dock onto TM sites (albeit different from each other; see Discussion). Interestingly, PROG adopted similar poses to dock onto two TM residues conserved between $\beta_1$ and $\beta_4$: Tyr32 and Phe33 in TM1 for $\beta_1$ and $\beta_4$, respectively, and Trp163 and Trp176 in TM2 for $\beta_1$ and $\beta_4$, respectively. In both subunits, a single PROG molecule is able to cover the area between the two TM domains and thus "bridge" TM1 and TM2 (Fig. 2a, b). To determine the relevance of the identified residues to PROG activation of $\beta_1$-containing channels, we introduced non-

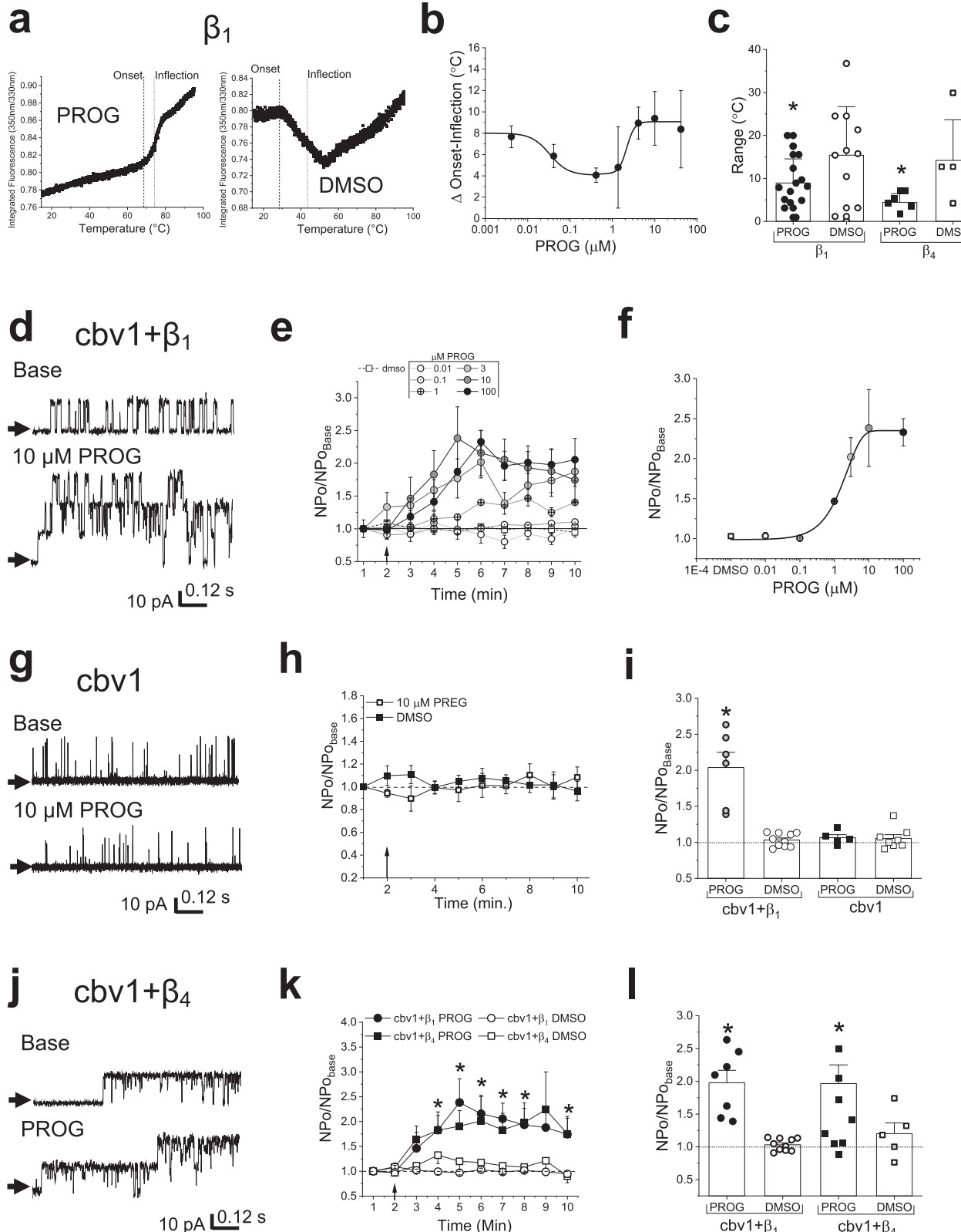

conserved, point mutations: Y32V and W163I in TM1 and TM2, respectively. After reconstitution of slo1 in the presence of either $\beta_1$ point mutant, the maximal-effective concentration of PROG (10 µM) was added to the *cis* bilayer chamber and the resulting channel activity was compared to that in the presence of vehicle control (DMSO). Either substitution effectively abolished PROG action (Fig. 2c–f), indicating that Tyr32 and Trp163 are *both* required for PROG recognition and eventual channel activation. Importantly, other mutations introduced

(negative controls: Y32F and Y42V) were shown to still bind PROG and be significantly activated by 10 µM PROG when compared to vehicle control (DMSO; Supplementary Fig. 3a, b). It is noteworthy that all $\beta_1$ subunit mutations we introduced resulted in $\beta_1$ constructs that functionally coupled to the BK channel-forming subunits when reconstituted into planar lipid bilayers; as found for their wild type (wt) counterparts, association of these mutants to slo1 resulted in a significant increase in NPo and lengthening of mean open times when

**Fig. 1 | Progesterone binds to and activates the BK channel complex in a concentration-dependent manner through the $\beta_1$ regulatory subunit, in absence of the native cellular environment. a** Original trace records of the thermal unfolding of $\beta_1$ in the presence of PROG (10 μM; left) and the vehicle control (DMSO; right). **b** Graphical depiction of the averaged PROG-driven, concentration-dependent changes in $\beta_1$ thermal unfolding, measured by the delta (Δ) of onset to inflection point. Averaged data fitted to a Hill coefficient function for low (0.004-0.4 μM) and high concentrations (0.4-40 μM). Data are shown as mean ± SEM; $n$ = 18, 13, 7 and 5 for each group, respectively. **c** Averaged changes in $\beta_1$ and $\beta_4$ thermal unfolding produced by 10 μM PROG and the vehicle control (DMSO), measured by the delta (Δ) of onset to inflection point. Data are shown as mean ± SEM; geometric shapes indicate individual records. *Statistically significant from vehicle control ($P$ = 0.047; $P$ = 0.0143, for $\beta_1$ and $\beta_4$, respectively; two-sided Mann–Whitney $U$-test; $n$ = 18, 13, 7 and 5 for each group, respectively). **d** Representative channel recordings of slo1 (cbv1 isoform)+$\beta_1$ after incorporation into POPE/POPS (3:1 w/w) artificial lipid bilayers, comparing the basal channel activity (top) and channel activity after 10 μM PROG was added to the *cis*-chamber (bottom). Channel openings are shown as upward deflections; black arrows indicate the baseline, with no channel activity. **e** Averaged data showing the time-dependent changes in NPo/NPo$_{basal}$ for each minute of recording, comparing the concentration response curve (0.01 μM-100 μM) of PROG to the vehicle (DMSO) time-matched controls for cbv1 + $\beta_1$. A black arrow indicates the time at which PROG/DMSO was added. A horizontal dashed line at 1 ($y$-axis) indicates no change from baseline activity ($n$ = 10, 4, 4, 3, 7 and 3 for 0, 0.1, 1, 3, 10 and 100 μM PROG, respectively). **f** Averaged changes in NPo/NPo$_{basal}$ of the maximal cbv1 + $\beta_1$ channel activity after in artificial lipid bilayers produced by each concentration of PROG. A black line indicates exponential fitting of data to the obtain a concentration-response curve ($n$ = 10, 4, 4, 3, 7 and 3 for 0, 0.1, 1, 3, 10 and 100 μM PROG, respectively). **g** Original trace recordings of cbv1 before and after 10 μM PROG.

**h** Averaged data showing NPo/NPo$_{basal}$ for each minute of recording, comparing cbv1 channel activity in the presence of either PROG (10 μM) or the vehicle, time-matched controls. A black arrow indicates the time at which PREG/vehicle control was added. A horizontal dashed line at 1 ($y$-axis) indicates no change from baseline activity; $n$ = 7, 10, 5 and 8 for each group, respectively. **i** Average data comparing the maximal activity of cbv1 versus cbv1 + $\beta_1$ in the presence of either the vehicle control or 10 μM PROG. Geometric shapes indicate individual channel recordings. *Statistically significant from cbv1+PROG; mean ± SEM; two-sided Mann–Whitney $U$-test ($P$ = 0.041; $n$ = 7, 10, 5 and 8 for each group, respectively). **j** Representative slo1 (cbv1 isoform)+$\beta_4$ channel recordings, obtained after incorporation into POPE/POPS (3:1 w/w) bilayers comparing the basal channel activity (top) and channel activity after 10 μM PROG was added to the *cis*-chamber (bottom). Channel openings are shown as upward deflections; black arrows indicate the baseline, with no channel activity. **k** Averaged data showing NPo/NPo$_{basal}$ for each minute of recording, comparing 10 μM PROG to the vehicle (DMSO) time-matched controls for cbv1 + $\beta_1$ and cbv1 + $\beta_4$. A black arrow indicates the time at which PROG/DMSO was added. A horizontal dashed line at 1 ($y$-axis) indicates no change from baseline activity. Data are shown as mean ± SEM; $n$ = 7, 10, 8 and 5 for each group, respectively. *Statistically significant from the corresponding vehicle control ($P$ = 0.0439, 0.0480, 0.0419, 0.0219, 0.0178 and 0.0012 for cbv1 + $\beta_1$. $P$ = 0.041, 0.0402, 0.0449, 0.0378, 0.0236 and 0.008 for cbv1 + $\beta_4$; two-sided Mann–Whitney $U$-test). **l** Averaged maximal changes in channel activity from the presence of 10 μM PROG or the vehicle control for cbv1 + $\beta_1$ and cbv1 + $\beta_4$. Geometric shapes indicate individual records ($P$ = 0.0419 and $P$ = 0.0449 for cbv1 + $\beta_1$ and cbv1 + $\beta_4$, respectively; two-sided Mann–Whitney $U$-test). From (**d**) to (**l**), channel activity was recorded at 0 mV in the presence of 30 μM Ca$^{2+}$ in the *cis*- and *trans*-solutions. Data are shown as mean ± SEM; $n$ = 7, 10, 8 and 5 for each group, respectively. *Statistically significant from vehicle control.

compared to slo1 homomers under identical voltage and internal calcium levels (Supplementary Fig. 4).

## A bridge formation between the two transmembrane domains in BK $\beta_1$ is required for channel activation by μM progesterone

The requirement of two residues each located in a separate TM for PROG action on channel activity is most likely explained by their simultaneous occupancy by PROG. This could be achieved by a single PROG molecule binding to the two residues of relevance with the steroid spanning the two TMs, as shown with our model for PROG docking onto $\beta_1$ and $\beta_4$ (Fig. 2a, b). Alternatively, the same result could be explained by the involvement of two different steroid molecules each bound to one individual residue in a single TM, as exemplified in Fig. 3b with the docking of progesterone 3-(O-carboxymethyl)oxime (PROG-oxime; Fig. 3a, bottom). To distinguish between these two possibilities, we next investigated the effects of 10 μM PROG-oxime on slo1 + $\beta_1$ complexes. As shown in Fig. 3b (left panel), this close structural analog of PROG docks onto $\beta_1$ by reaching Tyr32 and Trp163 individually, which is corresponded by its actual binding to wt $\beta_1$ (Fig. 3b; right panel). PROG-oxime, however, was unable to remain stably bound forming a bridge between the two TMs, which is likely attributed to the bulky nature of the oxime group. Using conditions identical to those used to probe PROG, PROG-oxime action on channel activity did not differ from that evoked by vehicle control (Fig. 3c–e). This result indicates that the bridge between the TMs is critical for PROG to activate $\beta_1$-containing BK channels. Further supporting our bridge model, computational simulations for PROG interactions with $\beta_1$Y32F proteins show that PROG, as expected for a negative control mutation, still interacts with the low affinity binding site by bridging the two TMs in the vast majority of the trajectories (>66%) (Supplementary Fig. 3c).

## Progesterone binding to conserved regions of the $\beta_1$ subunit loop is required for BK channel activation by the steroid

Since the concentration-binding curve (CRC) in Fig. 1b shows not only one site ("low-affinity") that binds μM PROG (which finds its functional

validation in the electrophysiological results shown in Figs. 1–3), but also a higher affinity site that binds subμM PROG, we next began to investigate the structural bases of such distinction. Remarkably, the double-mutant $\beta_1$Y32V + W163I including both substitutions that disrupt the low-affinity site (Fig. 2c, f), was still destabilized by low concentrations of PROG (subμM) (Fig. 4a; $P$ = 0.039). This supports the idea that, indeed, there is a high-affinity binding site in $\beta_1$ that is separate from the lower affinity binding site previously identified by which a single PROG molecule bridges both TMs. In this case, there could be allosteric communication between the high- and the low-affinity sites. Notably, the double-mutant $\beta_1$Y32V + W163I, which includes both substitutions that on their own disrupt the low-affinity site, was not destabilized by high concentrations of PROG (4.1 μM) despite the expectation that at this PROG level, the high-affinity site would be saturated. We can speculate that this double mutation introduces other, yet to be identified, site(s) for PROG binding and/or conformational changes that oppose the conformational changes triggered by PROG interaction with the high-affinity site. Formal testing of these possibilities requires future work involving scanning mutagenesis of amino acids that were not probed in this study. Since the low-affinity binding site required aromatic amino acids, we investigated similar amino acids within $\beta_1$ in search of the high-affinity site. Thus, we used computational modeling of the $\beta_1$ and $\beta_4$ subunits to dock PROG to *all atoms* of the protein structures and found that there was a conserved region of beta-pleated sheets *within the loop* of the $\beta_1$ and $\beta_4$ that docked PROG (Fig. 4b, c). Importantly, this site differs from the docking of PROG-oxime to the $\beta_1$ loop (Supplementary Fig. 6a; green) and utilizes another Trp for hydrogen bonding: Trp87 (Fig. 4b). Moreover, there was loss of PROG binding at both low and high concentrations (0.004 and 4.1 μM PROG, respectively) when Trp87 was mutated to Phe (Fig. 4a). However, when the loop of $\beta_4$ was attached to the $\beta_1$ TMs ($\beta_4$ loop in $\beta_1$ background chimera) 0.004 μM PROG still bound and destabilized the protein structure in MST (Fig. 4a; $P$ = 0.022).

In order to validate the loss of binding proposed by the docking model and MST, we investigated PROG-induced activation of α + $\beta_1$ channels after introducing the point mutation $\beta_1$W87F. After channel

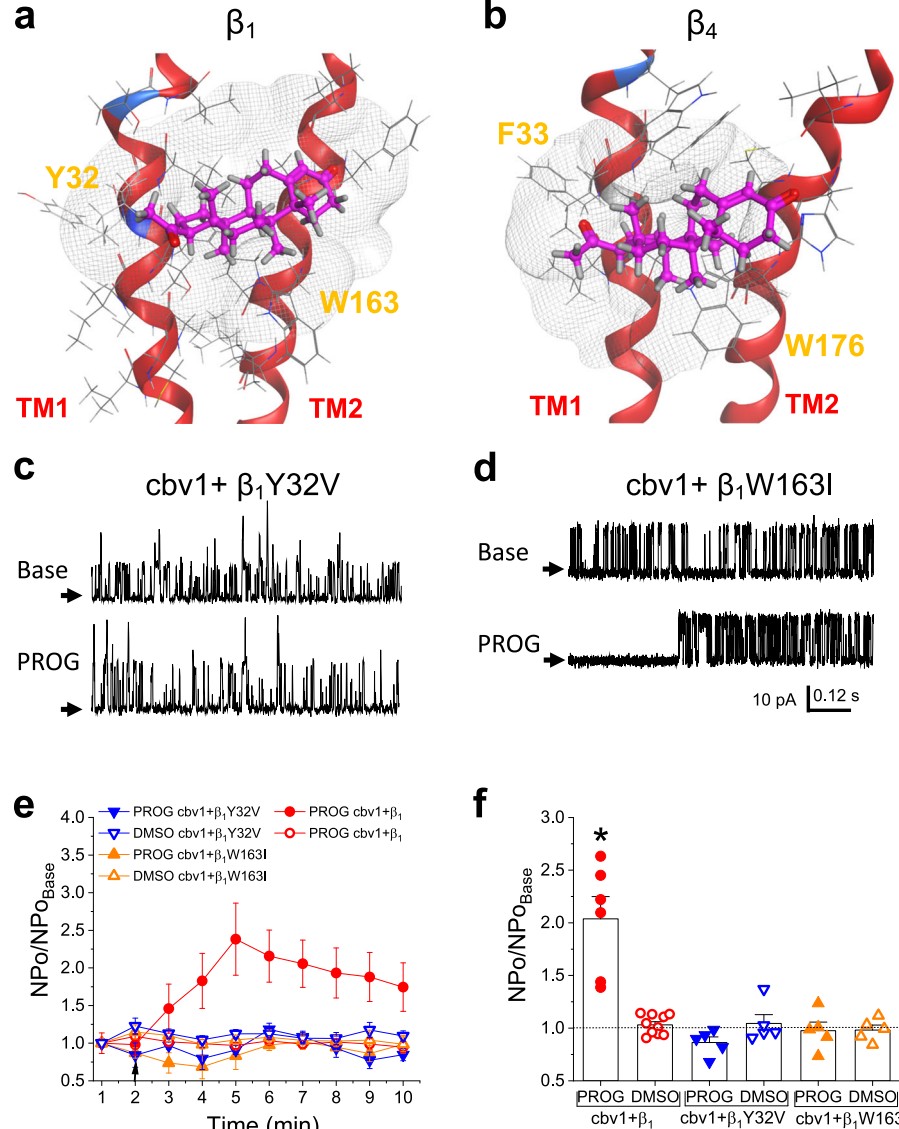

**Fig. 2 | Progesterone binds to conserved regions within the transmembrane domains of β1 and β4 to similarly activate BK channel complexes including either regulatory subunit. a** Representative snapshot of the most energetically favorable docking motif of PROG (pink) on the β1 regulatory subunit transmembrane domains. The theoretical binding pocket for PROG, depicting interactions with Tyr32 and Trp163. Here and in (**b**), gray netting reflects receptor surface map by van der Waals interactions. **b** Representative snapshot of the most energetically favorable docking motif of PROG (pink) on the β4 regulatory subunit transmembrane domains. The theoretical binding pocket for PROG, depicting interactions with Phe33 and Trp176. **c** Representative slo1 (cbv1 isoform)+β1Y32V channel recordings obtained after incorporation of cbv1 + β1Y32V into POPE/POPS (3:1 w/w) bilayers comparing the basal channel activity (top) and channel activity after 10 μM PROG was added to the *cis*-chamber (bottom). Channel openings are shown as upward deflections; black arrows indicate the baseline, with no channel activity.

**d** Representative slo1 (cbv1 isoform)+β1W163I channel recordings obtained after incorporation of cbv1 + β1W163I into POPE/POPS (3:1 w/w) bilayers comparing the basal channel activity (top) and channel activity after 10 μM PROG was added to the *cis*-chamber (bottom). Channel openings are shown as upward deflections; black arrows indicate the baseline, with no channel activity. **e** Averaged data showing $NPo/NPo_{basal}$ for each minute of recording, comparing 10 μM PROG to the vehicle (DMSO) time-matched controls. A black arrow indicates the time at which PROG/vehicle control was added. Data are shown as mean ± SEM; $n = 7, 10, 5, 5, 5$ and 5 for each group, respectively. **f** Averaged maximal changes in channel activity from the presence of 10 μM PROG or the vehicle control for cbv1 + β1, cbv1 + β1Y32V, and cbv1 + β1W163I. Geometric shapes indicate individual records. Data are shown as mean ± SEM; $n = 7, 10, 5, 5, 5$ and 5 for each group, respectively. *Statistically significant from vehicle control; two-sided Mann–Whitney $U$-test ($P = 0.041$).

reconstitution into lipid bilayers as used with wt β1-containing heteromers, the maximal-effective concentration of PROG (10 μM) was added to the *cis* bilayer chamber and the change in NPo was compared to that in the presence of vehicle. Remarkably, the W87F substitution effectively abolished PROG action (Fig. 4d–f). These data indicate that the loop (whether from β1 or β4) is involved in PROG binding. Moreover, Trp87 in the β1 loop is critical for PROG binding at low concentrations allowing Tyr32 and Trp163 to be recruited for PROG binding as a bridge in the low-affinity site with eventual PROG-induced conformational changes (Fig. 4) that result in channel activation (Figs. 1 and 2).

Next, we performed molecular dynamic simulations on the high- and low-affinity binding sites to determine the difference(s) in protein conformation and eventual contribution of the two sites to BK channel activation. When PROG is bound between the two TMs at Tyr32 and Trp163, dynamics show loose binding to Tyr32 and strong hydrogen bonding with Trp163 (Fig. 4g compared to Supplementary Fig. 5a). Interestingly, when PROG is bound to the beta-pleated sheets within the β1 loop, this ligand receives polar receptor-ligand contouring by Trp87, which causes a conformational change to the loop-TM connections. This causes the alpha-helices to loosen (shown by blue semi-

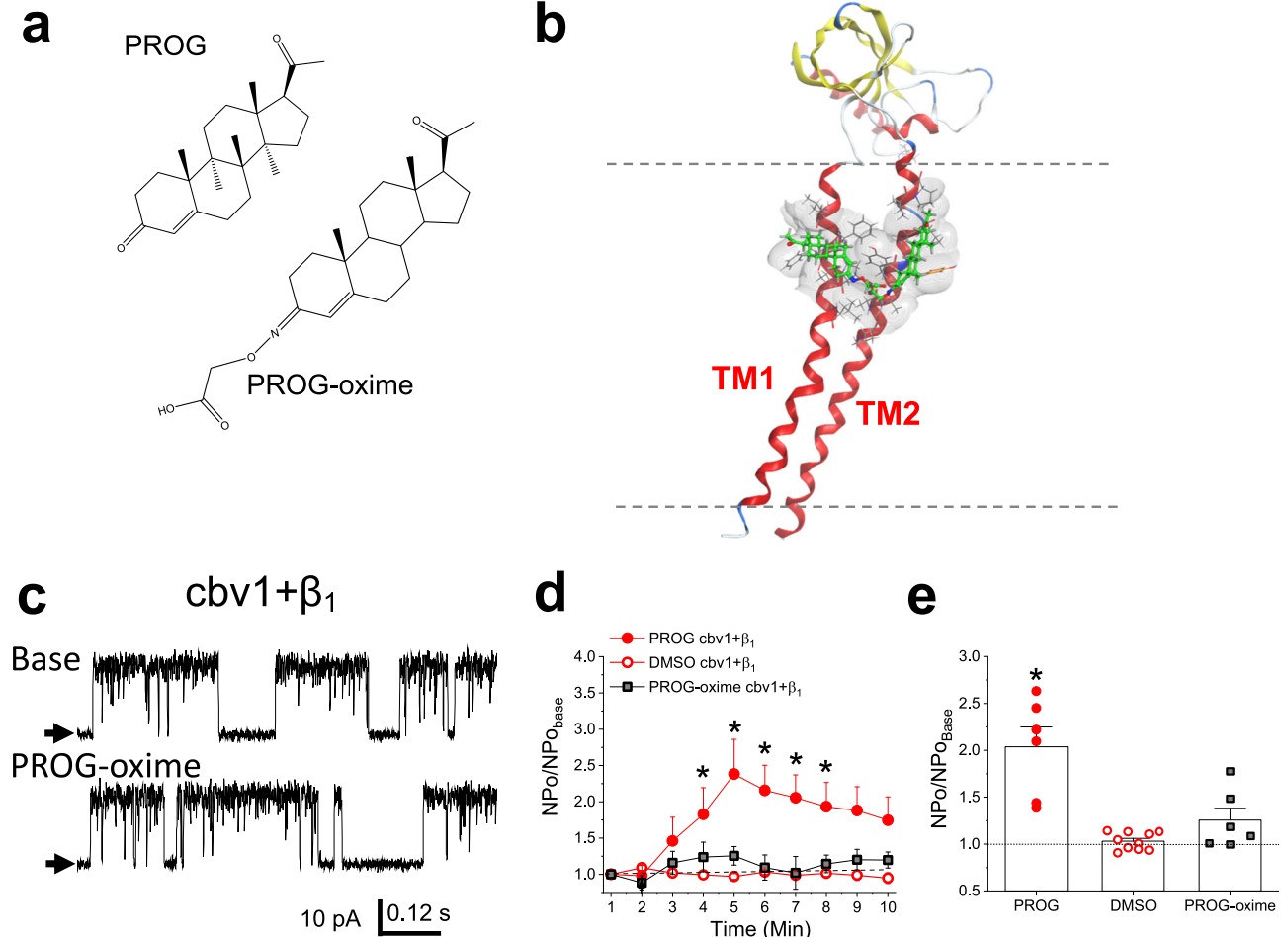

**Fig. 3 | Progesterone forms a bridge between amino acids Tyr32 and Trp163 from TM1 and TM2, respectively, and thus activates BK channels. a** Structural depictions comparing PROG (top) to progesterone 3-(O-carboxymethyl)oxime (PROG-oxime; bottom). **b** Representative snapshot of the most energetically favorable docking mode of PROG-oxime (green) on the $\beta_1$ regulatory subunit transmembrane domains on amino acids Tyr32 and Trp163. Plasma membrane is defined by dotted gray lines. Gray netting represents receptor surface maps by van der Waals interactions. **c** Representative slo1 (cbv1 isoform)+$\beta_1$ channel recordings obtained after incorporation of cbv1 + $\beta_1$ into POPE/POPS (3:1 w/w) bilayers comparing the basal channel activity (top) and channel activity after 10 μM PROG-oxime was added to the *cis*-chamber (bottom). Channel openings are shown as upward

deflections; black arrows indicate the baseline, with no channel activity. **d** Averaged data showing NPo/NPo$_{basal}$ for each minute of recording, comparing 10 μM PROG and 10 μM PROG-oxime to the vehicle (DMSO) time-matched controls for cbv1 + $\beta_1$. A black arrow indicates the time at which PROG/PROG-oxime/vehicle control was added. A horizontal dashed line at 1 (y-axis) indicates no change from baseline activity. Data are shown as mean ± SEM; $n = 7$, 10 and 6 for each group, respectively. **e** Averaged maximal changes in channel activity from the presence of 10 μM PROG, the vehicle control, or 10 μM PROG-oxime for cbv1 + $\beta_1$. Geometric shapes indicate individual records. Data are shown as mean ± SEM; $n = 7$, 10 and 6 for each group, respectively. *Statistically significant from vehicle control; two-sided Mann–Whitney *U*-test ($P = 0.041$).

loops) and move closer together (Fig. 4h compared to Supplementary Fig. 5b). This conformational change is mirrored when molecular dynamic simulations are run with PROG being docked to the loop (Fig. 4i and Supplementary Fig. 3d). Thus, the binding of PROG between the two TMs is enabled by the binding of PROG to the loop site, with the loosened alpha-helix of TM1 rotating Tyr32 for direct hydrogen bonding with one oxygen of PROG while retaining the oxygen-hydrogen bonding between PROG and Trp163 (Fig. 4i). Collectively, these data and electrophysiological results (Figs. 1d–f and 4d–f) indicate that while binding of PROG to the high-affinity site (Trp87) is required for the steroid to bind the low-affinity site, the former process does not directly translate into channel activation by PROG. Thus, the high-affinity site serves a permissive role whereas the low-affinity site serves as an effector of PROG binding to the loop.

**Progesterone activates vascular smooth muscle BK channels in their native membrane through their β₁ subunits**

Next, we addressed whether β₁-dependent, direct modulation of BK channels by PROG as identified via MST and functional studies where

BK heteromers were reconstituted into artificial binary lipid bilayers was sustained when the steroid was probed on vascular smooth muscle (SM) channels in their native membrane. Thus, we determined whether PROG modulated vascular SM BK channel complexes in *KCNMB1⁻/⁻* (β₁ KO) mice vs. their wt counterparts (C57BL/6J). Thus, based on our CRC data from recombinant BK channels (Fig. 1e,f), we chose to probe these native channels with PROG EC$_{100}$ (10 μM). Using the inside-out (I/O) configuration on SM cells isolated from middle cerebral artery (MCA) and bathed in solutions containing 30 μM free [Ca$^{2+}$], PROG reversibly increased BK channel activity in wt mouse by ~3-fold (Fig. 5a, c, d), a robust activation of similar magnitude to that obtained with recombinant channels. Consistent with the need for PROG to bind β₁ subunits to activate recombinant BK channels (Fig. 1g–i), BK channels from *KCNMB1⁻/⁻* mouse MCA SM cells were consistently resistant to PROG (Fig. 5b–d). Collectively, electrophysiological results from recombinant and native SM BK channels suggest that the different lipid composition and architecture between a planar two-phospholipid bilayer and those of a SM cell membrane do not critically modify PROG direct activation of β₁-containing BK channels.

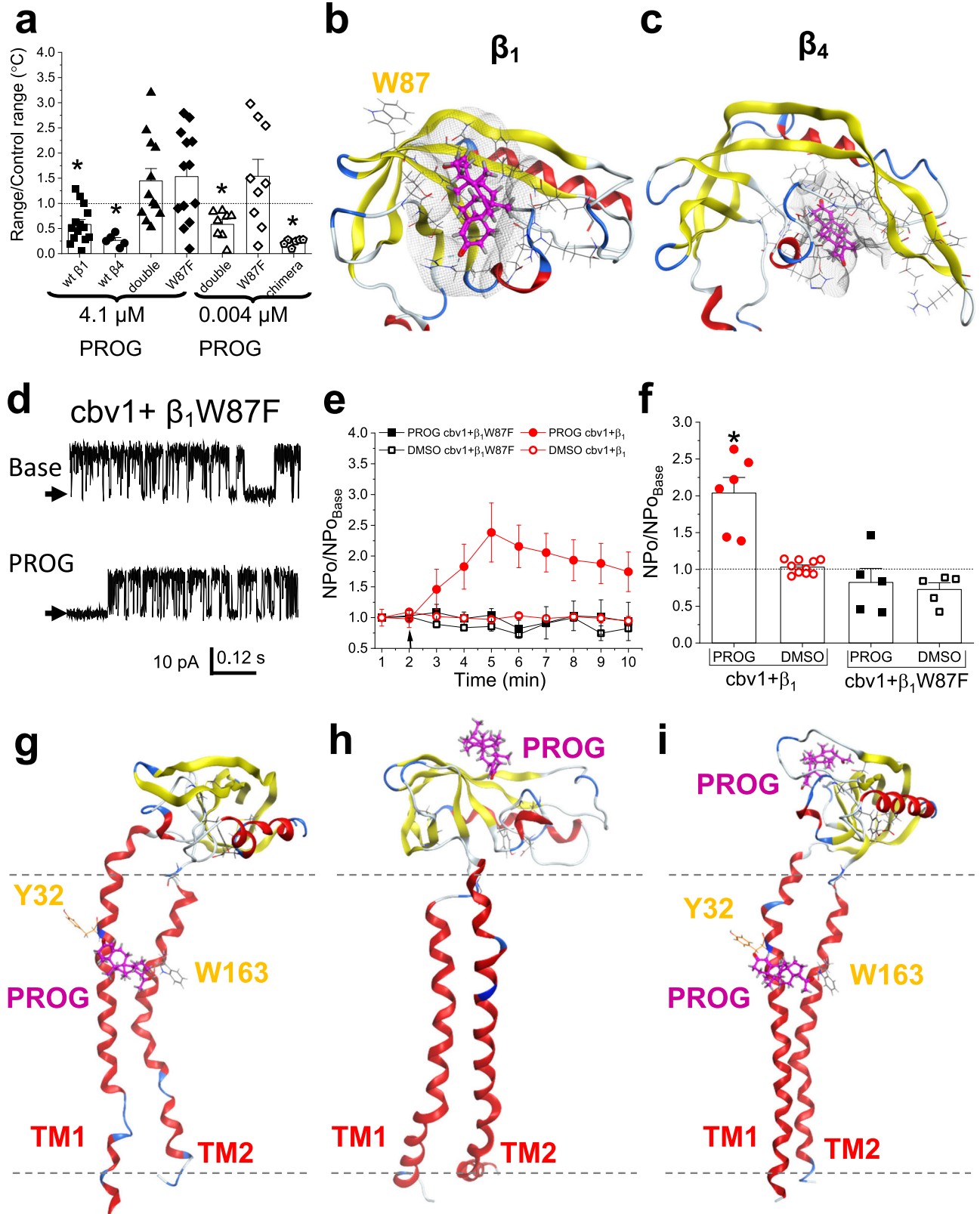

## Progesterone dilates cerebral arteries through smooth muscle BK channels, independently of circulating factors and endothelium

In order to determine whether activation of SM BK channels by PROG impacts organ, MCA segments were de-endothelialized, in vitro pressurized, and probed with the $EC_{100}$ (10 μM) PROG. We chose de-endothelialized MCA to remove any possible contribution of

endothelial factors and focus on a system where artery diameter is critically dependent on SM BK channel activity[15]. Viability of isolated MCA segments was determined by evaluating their contraction to 60 mM KCl-induced depolarization, which serves as reader of close-to-maximal, depolarization-driven MCA/brain artery constriction[30]. We first compared the vasoactive response of 10 μM PROG in presence and absence of paxilline, at a concentration that selectively blocks BK

**Fig. 4 | Progesterone binding to BK regulatory subunits involves two sites of high- and low-affinity located in conserved regions of $\beta_1$ and $\beta_4$ (loop and transmembrane domains, respectively) with the former serving a permissive role for the latter become effector of PROG action on channel function.**
**a** Averaged changes in $\beta_1$, $\beta_4$, $\beta_1$Y32V,W163I (double mutant), $\beta_1$W87F, and $\beta_1$-$\beta_4$ chimera (loop of $\beta_4$ with $\beta_1$ TMs) thermal unfolding produced by high (4.1 µM) and low (0.004 µM) concentrations of PROG measured by the delta (Δ) of onset to inflection point, normalized to the vehicle control (DMSO). Geometric shapes indicate individual records. Data are shown as mean ± SEM; $n$ = 18, 5, 12, 13, 8, 9 and 6 for each group, respectively. *Statistically significant from vehicle control ($P$ = 0.047, $P$ = 0.0143, $P$ = 0.039 and $P$ = 0.022, for $\beta_1$, $\beta_4$, double mutant and chimera, respectively; two-sided Mann−Whitney $U$-test). **b** Representative snapshot of the most energetically favorable docking motif of PROG (pink) on the $\beta_1$ regulatory subunit loop, with direct interactions with Trp87. **c** Representative snapshot of the most energetically favorable docking motif of PROG (pink) on the $\beta_4$ regulatory subunit loop. **d** Representative slo1 (cbv1 isoform)+$\beta_1$W87F mutant channel recordings obtained after incorporation of the channel complex into POPE/POPS (3:1 w/w) bilayers comparing the basal channel activity (top) and channel activity

after 10 µM PROG was added to the *cis*-chamber (bottom). Channel openings are shown as upward deflections; black arrows indicate the baseline, with no channel activity. **e** Averaged data showing NPo/NPo$_{base}$ for each minute of recording, comparing 10 µM PROG to the vehicle (DMSO) time-matched controls. A black arrow indicates the time at which PROG/vehicle control was added. Data are shown as mean ± SEM; $n$ = 7, 10, 5 and 5 for each group, respectively. **f** Averaged maximal changes in channel activity from the presence of 10 µM PROG or the vehicle control for cbv1 + $\beta_1$, cbv1 + $\beta_1$W87F. Geometric shapes indicate individual records. Data are shown as mean ± SEM; $n$ = 7, 10, 5 and 5 for each group, respectively. *Statistically significant from vehicle control; two-sided Mann−Whitney $U$-test ($P$ = 0.041). **g** Representative snapshot of the average final position of molecular dynamic simulations for the docking of PROG (gray) on the $\beta_1$ regulatory subunit TM. Here and in (**h, i**), plasma membrane is defined by dotted gray lines. **h** Representative snapshot of the average final position of molecular dynamic simulations for the docking of PROG (gray) on the $\beta_1$ regulatory subunit loop. **i** Representative snapshot of the average final position of molecular dynamic simulations for the docking of PROG (gray) on the $\beta_1$ regulatory subunit TM, with PROG pre-docked to the loop at amino acid Trp87.

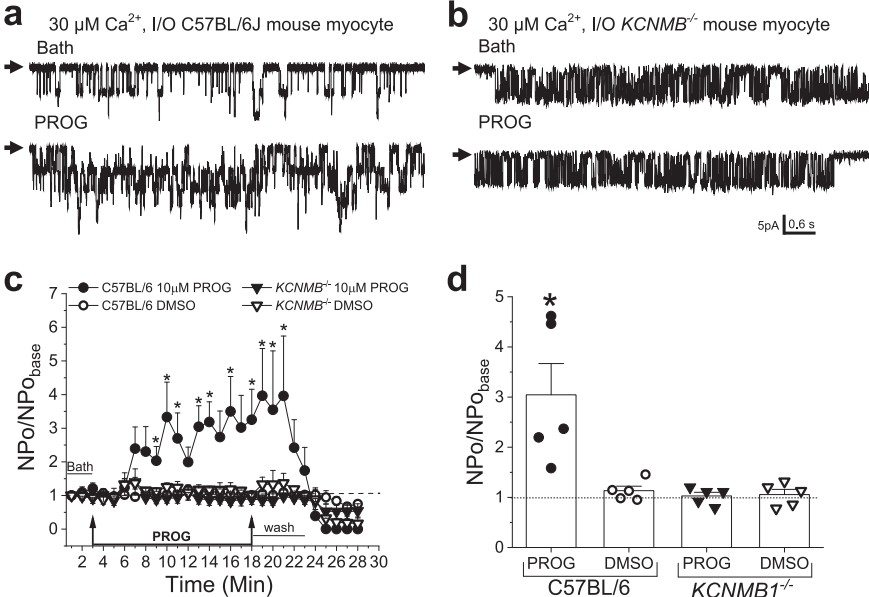

**Fig. 5 | Progesterone activation of BK channel activity by the $\beta_1$ regulatory subunit and the importance of the high-affinity site are sustained in native smooth muscle membranes.** **a** Example records of the activation of inside-out (I/O) patches of smooth muscle myocyte BK channel activity harvested from C57BL/6 mice before (top) and during (bottom) 10 µM PROG. Channel openings are shown as downward deflections; black arrows indicate the baseline with no channel activity. **b** Example records of the activation of inside-out patches of smooth muscle myocyte BK channel activity harvested from *KCNMB1*$^{-/-}$ mice before (top) and during (bottom) 10 µM PROG. All records were recorded in 30 µM free-Ca$^{2+}$ at 30 mV transmembrane voltage. Channel openings are shown as downward deflections; black arrows indicate the baseline with no channel activity. **c** Averaged

data showing NPo/NPo$_{basal}$ as a function of time, comparing channel activity in the presence of the vehicle control or PROG in either C57BL/6 or *KCNMB1*$^{-/-}$ mice. Black arrows indicate the time of change in perfusion (either bath perfusion or PROG perfusion). Data are shown as mean ± SEM; $n$ = 5 for each group. *Statistically significant from vehicle control ($P$ = 0.0425, 0.0366, 0.0426, 0.025, 0.0120, 0.0312, 0.0320, 0.0425, 0.0312 and 0.0425; two-sided Mann−Whitney $U$-test). **d** Averaged maximal effects of channel activity from C57BL/6 and *KCNMB1*$^{-/-}$ mice, in the presence of 10 µM PROG or the vehicle control. Geometric shapes indicate individual records. Data are shown as mean ± SEM; $n$ = 5 for each group. *Statistically significant from vehicle control ($P$ = 0.01208; two-sided Mann−Whitney $U$-test).

channels (1 µM[31–33]). PROG dilated MCA segments from male and female mice (Fig. 6a–d), this steroid being lost in the presence of 1 µM paxilline (Fig. 6c, d). This result indicates that PROG-induced dilation of MCA segments is mediated by BK channels. Moreover, comparison of male and female responses to PROG showed no difference between the sexes (Fig. 6e). Importantly, consistent with its failure to dock onto the low-affinity site of $\beta_1$ subunits (Fig. 3b) and activate $\beta_1$-containing BK channels (Fig. 3c, d, f), PROG-oxime failed to evoke MCA dilation (Fig. 6c, d, f). Lastly, comparing the effects of PROG-oxime in the presence and absence of 1 µM paxilline, revealed an MCA constriction that was preserved in the presence of the BK channel blocker (Fig. 6d, f), indicating that this action of PROG-oxime was due to extra-endothelial factors other than BK channels.

Importantly, in order to validate the biochemical and functional finding of the two PROG sites, we chose to permeabilize de-endothelialized MCA segments from male $\beta_1$ K/O mice with either wt $\beta_1$ cDNA, as positive control, or $\beta_1$ constructs encoding individual substitutions: W87F (located in the permissive, high-affinity site), Y32V and W163I (these two located in the effector, low-affinity site). Confirming at the organ level the roles of permissive and functional (effector) sites in PROG direct modulation of BK channels, 10 µM PROG failed to dilate MCA segments permeabilized with any of three constructs (Fig. 6g). Moreover, while Y32V and W163I substitutions ablated PROG-evoked MCA dilation, alcohol (ethanol; EtOH) constriction was still observed (Fig. 6g and Supplementary Fig. 7).

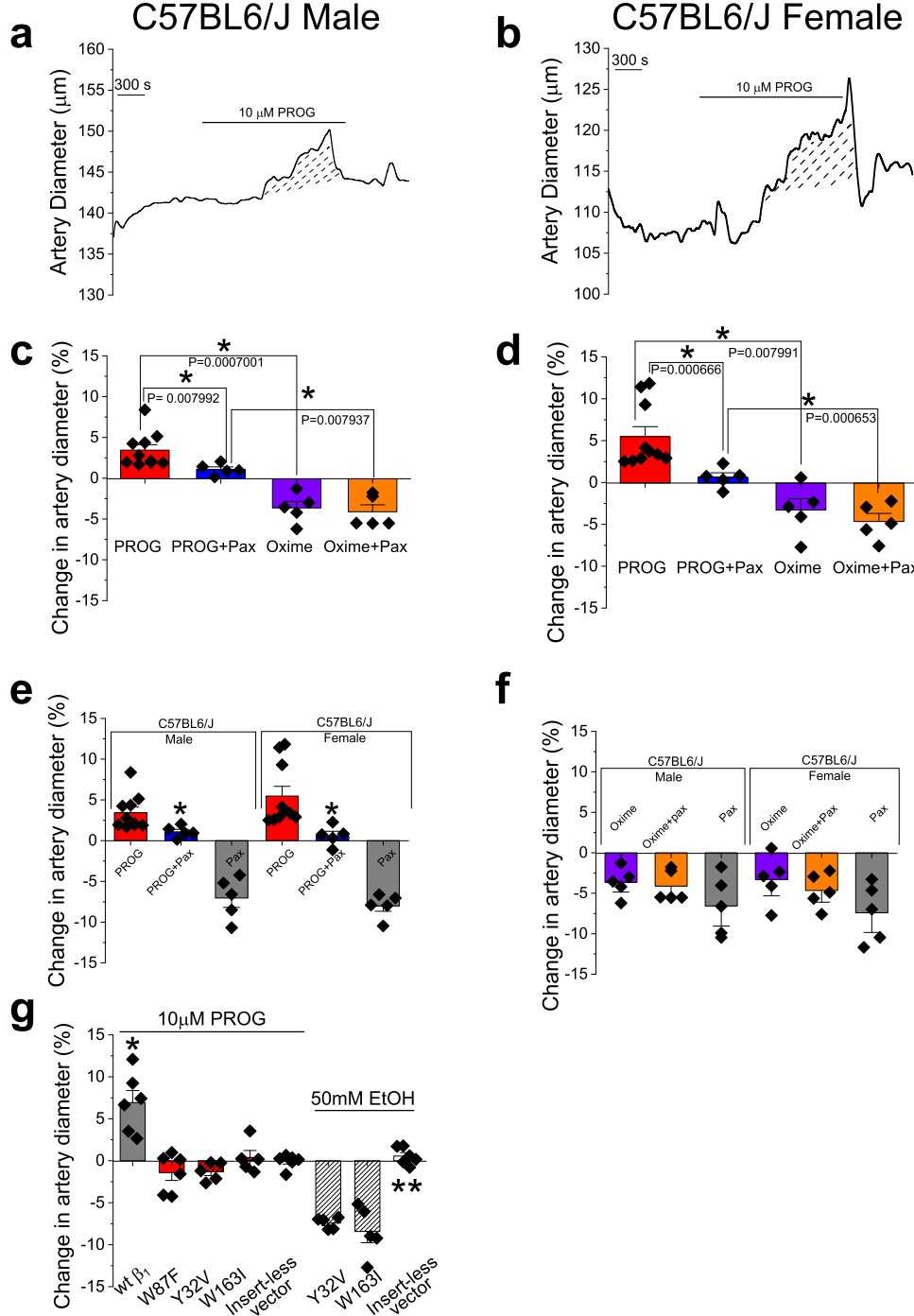

In order to evaluate PROG action on male and female MCA at the *organismal* level, we used a cranial window. This technique allowed us to continuously monitor the external diameter of resistance-size pial arteries that branch out of the MCA, as previously done to evaluate the alcohol pharmacology of these vessels[19,34,35]. For each experiment, baseline images of MCAs were captured prior to any drug application and used for reference of the fold-changes in MCA diameter throughout each experiment. All drugs were infused toward the cerebral circulation using an intra-carotid artery catheter as detailed in "Methods." Consistent with the in vitro data presented above, results in both sexes clearly show that PROG in vivo action on MCA branches (vasodilation) was drastically different than that of PROG-oxime (Fig. 7).

Lastly, to investigate the possibility of using PROG dilatory action in a clinically relevant scenario, we evaluated the effects of 10 μM PROG

on the CRC (10-75 mM) for EtOH-induced constriction of MCA. At these toxicologically relevant concentrations, EtOH is known to evoke constrictor of all branches of the Willis' circle, including MCA, in both rat and mouse[35,36], and cause brain hypoperfusion in humans[37]. Moreover, $\beta_1$ subunits are needed for this EtOH action[27] and agents that target this subunit, such as celastrol, have been demonstrated to counteract EtOH-induced MCA constriction[35]. Indeed, PROG blunted EtOH-induced vasoconstriction at all alcohol concentrations (Fig. 8a, b). Moreover, when the combination of 10 μM PROG and an EtOH concentration close to $EC_{100}$ (50 mM) was infused, the presence of PROG still protected against EtOH-induced cerebrovascular constriction (Fig. 8c-f). Thus, when taken together, our data from biochemical assays to organismal resolution, indicate that PROG dilates brain arteries in vitro and in vivo, similarly in male and female animals, through the activation of SM BK channel complex, an effect that is

**Fig. 6 | Consistent with our docking models, progesterone but not PROG-oxime, dilates cerebral arteries of male and female mice through the activation of β₁-containing BK channels. a** Diameter trace showing response to the application of 10 μM PROG in de-endothelialized, pressurized MCA segments of C57BL/6 male mouse. Diagonal dashed lines highlight response (area under the curve) to 10 μM PROG. **b** Diameter trace showing response to the application of 10 μM PROG in de-endothelialized, pressurized MCA segments of C57BL/6 female mouse. Diagonal dashed lines highlight response (area under the curve) to 10 μM PROG. **c** Comparison of percent changes in de-endothelialized MCA segments from C57BL/6 male mice in response to 10 μM PROG, 10 μM PROG in the presence of 1 μM paxilline, 10 μM progesterone 3-(O-carboxymethyl)oxime (PROG-oxime), and PROG-oxime in the presence of 1 μM paxilline. Geometric shapes indicate individual arterial records. Data are shown as mean ± SEM; $n = 9, 5, 5$ and 5 for each group, respectively. *Statistically significant ($P = 0.0079, 0.007$ and 0.0079; two-sided Mann–Whitney $U$-test). Here and in (**d**–**g**), each data point was obtained from a separate arterial segment. **d** Comparison of percent changes in de-endothelialized MCA segments from C57BL/6 female mice in response to 10 μM PROG, 10 μM PROG in the presence of 1 μM paxilline, 10 μM PROG-oxime, and 10 μM PROG-oxime in the presence of 1 μM paxilline. Geometric shapes indicate individual arterial records;

$n = 5$–10 for each group. Data are shown as mean ± SEM; $n = 10, 5, 5$ and 5 for each group, respectively. *Statistically significant ($P = 0.0006, 0.0079$ and 0.00065; two-sided Mann–Whitney $U$-test). **e** Comparison of percent changes in de-endothelialized MCA segments from male and female C57BL/6 J mice in response to 10 μM PROG, 10 μM PROG in the presence of 1 μM paxilline, and 1 μM paxilline. Geometric shapes indicate individual arterial records; $n = 9, 10, 5$ and 5 for each group, respectively. Data are shown as mean ± SEM. *Statistically significant ($P = 0.00799$ and $P = 0.00066$; two-sided Mann–Whitney $U$-test). **f** Comparison of percent changes in de-endothelialized MCA segments from male and female C57BL/6 J mice in response to 10 μM PROG-oxime, 10 μM PROG-oxime in the presence of 1 μM paxilline, and 1 μM paxilline. Geometric shapes indicate individual arterial records; $n = 5$ for each group. Data are shown as mean ± SEM. **g** Comparison of percent changes in de-endothelialized MCA segments in response to 10 μM PROG from male $KCNMB1^{-/-}$ KO mice permeabilized with β₁, β₁W87F, β₁Y32V, or β₁W163I mutant constructs. Geometric shapes indicate individual arterial records; $n = 6$ for each group. Data are shown as mean ± SEM. *Statistically different from insert-less vector ($P = 0.0035$; two-sided Mann–Whitney $U$-test); **Statistically different from insert-containing vector ($P = 0.0005$; two-sided Mann–Whitney $U$-test).

---

mediated by the channel β₁ regulatory subunit through a two-state binding model of high- and low-affinity sites (loop and TM domains, respectively).

## Discussion

Our study demonstrates that PROG binds to the β₁ subunit to activate the SM BK channel complex (Figs. 1 and 4). Moreover, this study identifies two unique steroid–protein interactions necessary for the biochemical binding and ion channel modulation by PROG. One is a high-affinity site; upon PROG binding there is no change in channel activity, yet this site is necessary for enabling PROG occupancy of the low-affinity site, underscoring its permissive role (Figs. 1f, 4, and 6g). The second site while showing a lower affinity for PROG, is contributed by the two TMs with a single PROG molecule forming a bridge between the two TMs; this binding is critical for PROG activation of the BK channel complex (Figs. 1–4). Thus, these data provide a model of steroid-BK channel interactions which is substantially more complex than previously identified single sites within a single TM domain for cholanes[16], or 17β-estradiol[18]. Our model also underscores a possible dissociation between binding and functional data when evaluating steroid action on ion channels, as the high-affinity PROG site just serves as a permissive role for PROG-induced activation upon steroid binding to the low-affinity site (Figs. 1f, 4, and 6g). Lastly, our study demonstrates that PROG direct action on β₁-containing BK channels through two recognition sites of different affinity contributes to dilation of MCA both in vitro and in vivo, and counteracts the cerebrovascular constriction evoked by intoxicating levels of EtOH.

Our current data documenting that PROG activates SM BK channels by binding directly to the β₁ subunit (Figs. 1–4) drastically differ from our recent study on BK channel modulation by pregnenolone. This neurosteroid, which may serve as PROG precursor, inhibits BK channels through recognition by the channel-forming α subunits, with β₁ presence showing no drastic impact on pregnenolone action[19]. It is noteworthy that BK channel inhibition via α subunits is shared by pregnenolone's precursor, cholesterol[19,20,38]. Thus, there seems to be a clear pattern of BK channel modulation by steroids: cholesterol and pregnenolone reduce BK channel activity, an action for which the expression of channel-forming α subunits is sufficient[19,20]. On the other hand, PROG (current study), 17β-estradiol[18,39,40] and bile acids and related cholanes[16,17,26,29] increase channel activity, an action that requires regulatory β subunits.

There are, however, significant differences in the structural bases of β subunit-supported activation of BK channels by steroids. Regarding cholanes, activation of BK channels and site-recognition by steroids requires accommodation of a bean-shaped molecule with

binding to Thr169 being critical[17,26,39]. This residue, present in β₁ is not conserved in the other β types, making cholane activation of BK channels "β₁-selective"[16] and thus, relevant to tissues that highly express this subunit such as vascular SM[13,14]. In turn, while PROG and 17β-estradiol both activate BK channels via β₁ subunit involving a common residue, Trp163[18] (Figs. 2 and 4), channel activation by PROG uniquely requires one steroid molecule making a "bridge" between the two TMs (Figs. 2, 3, and 4I). The vertical position of 17β-estradiol between TM1 and TM2, sitting between Trp163 and Phe166[18], is rather similar to the positioning of PROG-oxime (Fig. 3b). However, a significant difference between PROG-oxime (ineffective channel activator; Figs. 3, 6, and 7) and 17β-estradiol (effective) is that the former lacks the binding to Phe166 whereas both Trp163 and Phe166 are required for 17β-estradiol binding and activation of β₁-containing channels. The differences between 17β-estradiol and PROG docking onto β₁ could be due to either or a combination of the two following reasons: (1) PROG is more hydrophobic and therefore more reactive than the hydroxyl groups found on either ends of 17β-estradiol, allowing PROG to locate horizontally in the hydrophobic core of the bilayer; (2) PROG binding to the high-affinity site within the loop (on which 17β-estradiol cannot dock) primes the TMs for the formation of the PROG bridge (Fig. 4g–I and Supplementary Fig. 5). It is important to note, however, that the sites in β₁ TM2 for these two β₁-mediated activating steroids do not fully overlap: the hydrogen bonding between 17β-estradiol is mainly with Phe166[18] and not Trp163, as found here with PROG (Figs. 2 and 4).

How PROG binding to β₁ translates into channel activation is speculative at this point. Previous molecular modeling results showed that the two TMs of β₁ resided between S0 and S1 of two α subunits[18]. While the PROG high-affinity site does not translate into channel activation (Figs. 1f, 4d–f, and 6g), PROG binding to both sites seems to stabilize the β₁ TMs in a configuration (Fig. 4i) that is tighter than the configuration adopted when PROG is solely bound to the low-affinity site (Figs. 2a and 4g). Indeed, PROG docking to the β₁ loop seems to prime TM1 by loosening the α-helix coil, which rotates Tyr32 to point in the same direction as Trp163 (Fig. 4h, i). With Tyr32 and Trp163 in the same direction, the oxygens at either end of PROG now can form hydrogen bonds with the hydrophobic amino acids, thus stabilizing and pulling the TMs closer together (Fig. 4i). Based on a cryoEM structure that includes four β₄ subunits along with their α counterparts in a BK channel heteromeric complex[41], computational modeling shows that PROG binging to its high-affinity site still occurs at the loop, a model that highly likely applies to β₁ as well (Supplementary Fig. 8). The stoichiometry of PROG binding to cerebrovascular BK channels, however, cannot be established at this point; it still remains unknown

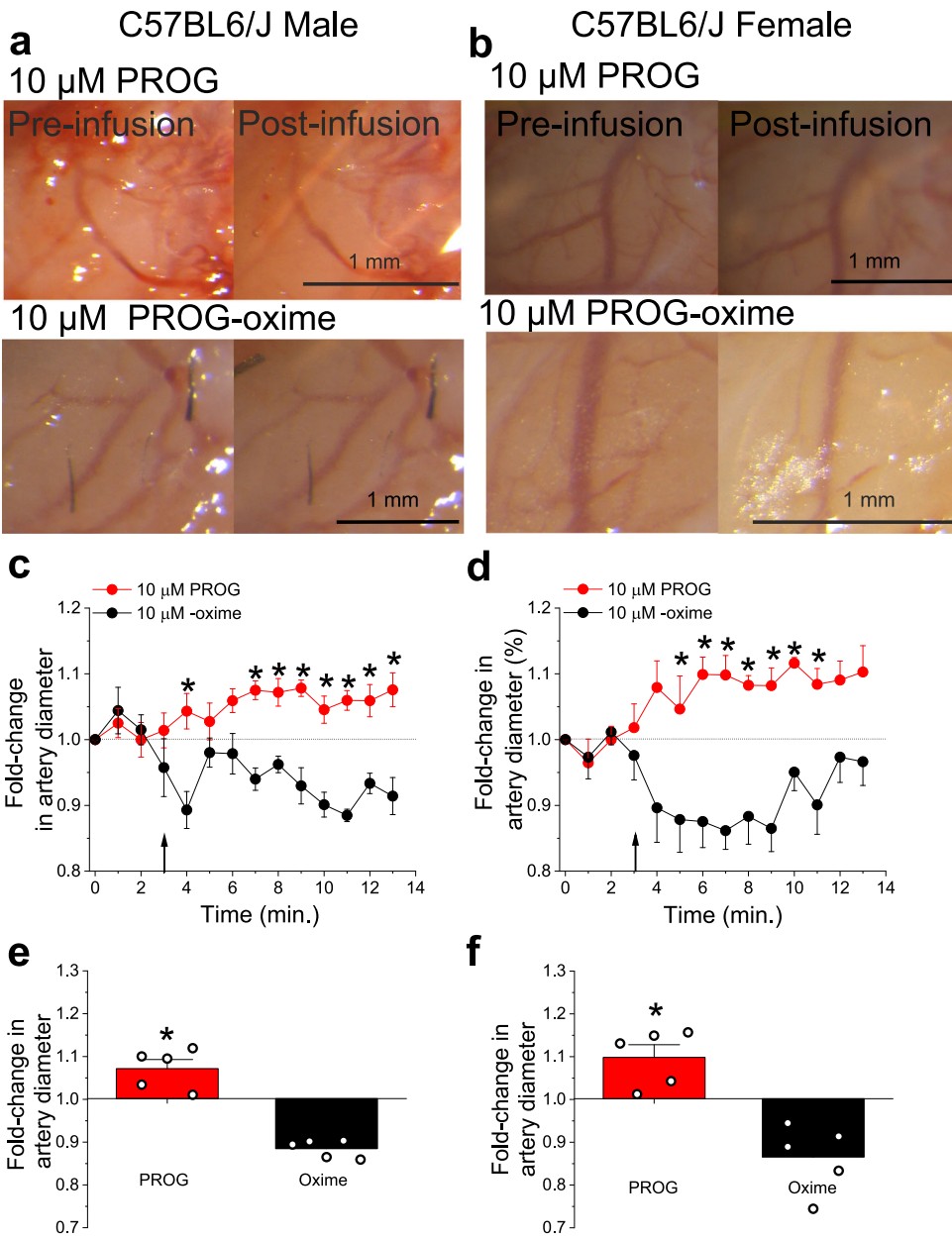

**Fig. 7 | Through β₁-involvement, progesterone similarly dilates cerebral arteries of male and female mice when evaluated in vivo. a** Cranial window images of C57BL/6J male MCA before any infusion (pre-infusion) and after infusion of 10 µM PROG or 10 µM PROG-oxime via the carotid artery. **b** Cranial window images of C57BL/6J female MCA before any infusion (pre-infusion) and after infusion of 10 µM PROG or 10 µM PROG-oxime via the carotid artery. **c** Averaged fold-changes in male MCA diameter comparing the responses to 10 µM PROG and 10 µM PROG-oxime for each minute of recording compared to baseline diameter determined from the image taken immediately prior to any infusion. A horizontal dashed line from 1 (y-axis), underscores no change in artery diameter. A black arrow indicates the beginning of infusion. Number of data points (n)=5 for each group, each data point was recorded from a separate animal. Data are shown as mean ± SEM. *Statistically significant from vehicle control ($P = 0.0158, 0.0079, 0.0079, 0.007, 0.0111, 0.0119, 0.011$ and $0.007$; two-sided Mann–Whitney U-test). **d** Averaged fold-changes in female MCA diameter comparing the responses to 10 µM PROG and 10 µM PROG-oxime for each minute of recording compared to baseline diameter determined from the image taken immediately prior to any infusion. A horizontal dashed line from 1 (y-axis), underscores no change in artery diameter. A black arrow indicates the beginning of infusion. Number of data points (n)=5 for each group, each data point was recorded from a separate animal. Data are shown as mean ± SEM. *Statistically significant from vehicle control ($P = 0.0095, 0.0119, 0.007, 0.0079, 0.0079, 0.0063$ and $0.0069$; two-sided Mann–Whitney U-test). **e** Averaged percentage of maximal artery changes in vivo produced by PROG and PROG-oxime in male C57BL/6 mice. Data are shown as mean ± SEM. Geometric shapes indicate individual records. *Statistically significant from vehicle control ($P = 0.012$; two-sided Mann–Whitney U-test). **f** Averaged percentage of maximal artery changes in vivo produced by PROG and PROG-oxime in female C57BL/6 mice. Data are shown as mean ± SEM. Geometric shapes indicate individual records. *Statistically significant from vehicle control ($P = 0.007$; two-sided Mann–Whitney U-test).

the actual stoichiometry of α and β₁ subunits in these native channels. Our model shows that while PROG binds between defined structural elements of individual β₄, subunits, subunit interfaces are enriched with disordered areas of the neighboring β loop (Supplementary Fig. 8). These disordered areas are expected to have high degree of spatial flexibility, thus likely favoring the transduction of the conformational changes triggered by PROG binding to β subunits into modification of BK channel gating and activity. This speculative series of changes would not occur with PROG-oxime: the most energetically favorable docking models of PROG-oxime do not affect the distance

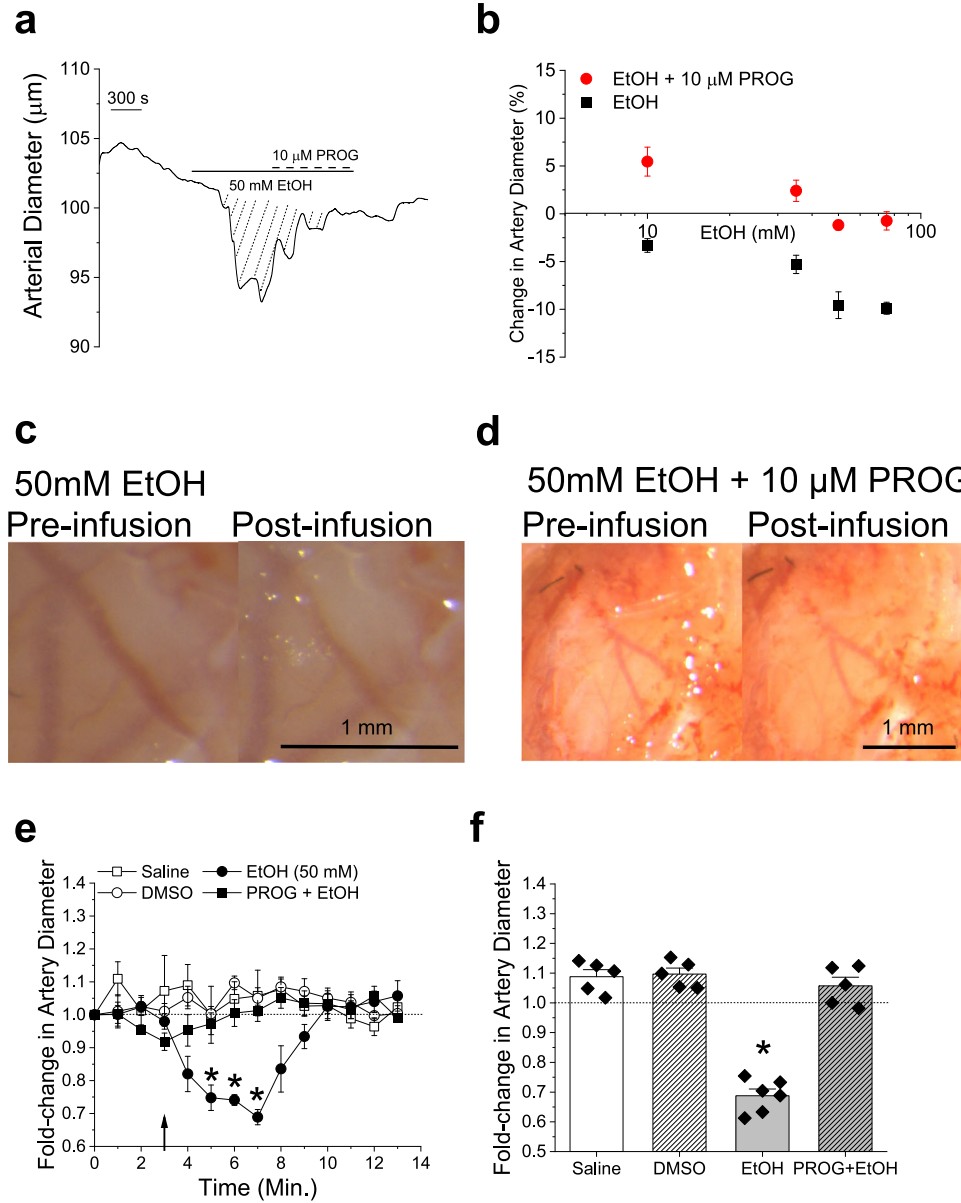

**Fig. 8 | Progesterone blunts ethanol-induced constriction of cerebral arteries both in vitro and in vivo. a** Diameter trace showing response to the application of 50 mM ethanol (EtOH) followed by 50 mM EtOH in the presence of 10 μM PROG in de-endothelialized, pressurized MCA segments of C57BL/6 male mouse. Diagonal dashed lines highlight response (area under the curve) to 50 mM EtOH. **b** Comparison of percent changes in de-endothelialized MCA segments from C57BL/6 male mice in response to 10–75 mM EtOH in the presence and absence of 10 μM PROG. Data are shown as mean ± SEM; $n = 5$ for each group. **c** Cranial window images of C57BL/6J male MCA before any infusion (pre-infusion) and after infusion of 50 mM EtOH via the carotid artery. **d** Cranial window images of C57BL/6J male MCA before any infusion (pre-infusion) and after infusion of 50 mM EtOH, combined with 10 μM PROG, via the carotid artery. **e** Averaged fold-changes in female

MCA diameter comparing the responses to 50 mM EtOH and 50 mM EtOH combined with 10 μM PROG, for each minute of recording compared to baseline diameter determined from the image taken immediately prior to any infusion. A horizontal dashed line from 1 (*y*-axis), underscores no change in artery diameter. A black arrow indicates the beginning of infusion. Number of data points (n)=5 for each group, each data point was recorded from a separate animal. Data are shown as mean ± SEM. *Statistically significant from vehicle control ($P = 0.0312$, 0.021 and 0.0081; two-sided Mann–Whitney *U*-test). **f** Averaged percentage of maximal artery changes in vivo produced by EtOH and PROG in male C57BL/6 mice. Data are shown as mean ± SEM; $n = 5$ for each group. Geometric shapes indicate individual records. *Statistically significant from vehicle control ($P = 0.0081$; two-sided Mann–Whitney *U*-test).

between the β₁ TMs (Fig. 3). Moreover, molecular dynamics simulations of the PROG-oxime bridge show that this steroid, when compared to PROG, moves the Tyr32 side chain relative to TM1, thus disfavoring our hypothesized, conformational domino-effect (Supplementary Fig. 6) and leading to its inefficacy to modulate channel activity.

Lastly, it is interesting that the PROG high-affinity site identified here provides binding events at natural PROG concentrations found within blood serum[42] (0.01–1 μM). However, the functional activation

of the BK channel complex occurs at therapeutic concentrations of PROG[24,28,43,44]. Both the region within the beta-pleated sheets for high-affinity binding to the loop as well as the low-affinity Tyr32, Trp163 TM site are conserved across *all* regulatory β subunits (Figs. 2 and 4) and thus, the mechanism proposed in our study might be applied to previous data showing that μM PROG can activate β₂- or β₄-containing BK channels[25]. Given that the vasoactive properties of PROG are being investigated for their therapeutic implications[43,44], our present results documenting that cerebrovascular dilation by PROG is largely

independent of circulating, metabolic and endothelial factors but mediated via steroid binding sites in BK $\beta_1$, and the conservation across all $\beta$ subunits[13,14] of the unveiled binding sites, it is critical that future studies investigate the impact of direct modulation of BK channel by PROG on organs that express subunits other than $\beta_1$[13,14]. Additionally, given the similarities between 17$\beta$-estradiol[18] and PROG (current data) sites, future studies should look at the possible competition between these two steroids for the low-affinity site and activation of BK channel complexes.

In conclusion, this study shows that PROG activates BK channels through direct recognition by their $\beta_1$ subunit. While additional binding sites to be discovered cannot be rule out, we propose that a unique interaction between PROG and $\beta_1$ results from sequential binding to two sites. First, a permissive, yet non-functional, high-affinity site within the $\beta_1$ loop involves Trp87. Upon this permissive binding, PROG bridges $\beta_1$ TMs at the low-affinity site and exerts its activatory effect on BK channels. Therefore, the two binding sites elucidated by this study through electrophysiological, molecular and biochemical evaluation, provide a glimpse at a mechanism in steroid-ion channel recognition, which clearly differs from previous, simpler models unveiled for other steroids[16–18,29]. The limitations of our study will be overcome by additional work to determine the contributions to PROG-BK channel interactions from (a) different membrane lipid species, (b) additional amino acids within $\beta_1$ subunits, and (c) channel inter-subunit crosstalk that might participate in PROG recognition and/or transduction of binding events into BK channel gating. At the organ level, our data and model for the direct binding of PROG to BK channel complexes allow for a better understanding of possible pharmacological effects of steroids via direct ion channel interaction, and the eventual identification of synthetic analogs that could mimic or antagonize PROG cerebrovascular effects through such direct interaction. Thus, these agents could be pursued as future therapeutic options for prevalent neurological and vascular disorders[24,43,44].

## Methods

### Ethical aspects of research
The care of animals and experimental protocols were reviewed and approved by the Institutional Animal Care and Use Committee of the University of Tennessee Health Science Center, which is an institution accredited by the Association for Assessment and Accreditation of Laboratory Animal Care international (AAALACi).

### Protein preparation
BK channel-forming slo1 cDNA (cbv1 isoform; AY330293) was cloned from rat cerebral artery SM cells[4], and inserted into pcDNA3.1 vector as described[20]. BK $\beta_1$ cDNA (*KCNMB1* product; NP_011452; FJ154955) were isolated from rat cerebral artery SM cells and cloned as described[29]. $\beta_4$ cDNA (*KCNMB4*; NP_055320; AY028605) were generously provided by Dr. Ligia Toro. CHO cells were transiently transfected with cbv1 $\pm$ $\beta_1$-, or $\beta_4$-carrying pcDNA3.1 using X-tremeGENE HP (Sigma-Aldrich). Mutations to $\beta_{1/4}$ cDNAs were conducted by Genscript, with sequences being confirmed to be >99% identical to *KCNMB1* product; NP_011452 or $\beta_4$ *KCNMB4*; NP_055320; respectively. $\beta_{1/4}$ and mutant constructs were flag-tagged (DYKDDDDK-tag) by Genscript services for subsequent immunoprecipitation and use in microscale thermophoresis studies.

Proteins for electrophysiological recordings in artificial lipid bilayers remained untagged. Cells were transiently transfected as described above, pelleted and re-suspended on ice in 10 ml of buffer solution of the following composition (mM): 30 KCl, 2 MgCl$_2$, 10 HEPES, 5 EGTA; pH 7.2. A membrane preparation was obtained using a sucrose gradient as described[45]. Aliquots were stored at −80 °C. BK channel-mediated ionic current was recorded in planar lipid bilayers consisting of a POPE:POPS 3:1 (w/w) mixture. Lipid mixtures were dried under N$_2$ and re-suspended in 25 mg/ml of n-decane BK currents

following incorporation into horizontal bilayers at 300/30 mM *cis/trans* gradient as detailed elsewhere[17]. Ion currents were obtained during a 10-min gap-free recording at 0 mV in the presence of 0.01, 10 µM PROG, 10 µM PROG-oxime, or vehicle control (dimethyl sulfoxide; DMSO). Currents were obtained in gap-free mode using a Warner BC-525D amplifier, low pass-filtered at 1 kHz and sampled at 5 kHz with Digidata 1550B/pCLAMP 11 (Molecular Devices). For comparison with previous data from us[16,17,45–47], studies were conducted at 20–22 °C. PROG stock, oxime stock or vehicle control (DMSO) was added to the bilayer *cis* chamber to reach final concentrations (0.01–10 µM), with all compounds being dissolved with the same concentration of DMSO (0.1%); $n = 5$–10 for each group.

### Immunoprecipitation and microscale thermophoresis
Wt $\beta_1$, wt $\beta_4$ and mutant constructs were purified by immunoprecipitation using Dynabeads Protein G Kit (10007D by Invitrogen) and following the manufacturer's instructions. Rabbit polyclonal anti-flag antibody (10 µg; Abcam, Cat# ab205606, RRID:AB_2916341) was cross-linked with dynabeads. Transfected CHO cell lysate supernatant was divided into aliquots containing 20$^7$ cells per column. Following precipitation, kit instructions for non-denaturing elutions were followed; one elution of 30 µl was collected for each column, and pH was adjusted to 7.5 using UltraPure 1 M Tris-HCl buffer pH 7.5 (cat# 15567027 by Fisher Scientific). PROG stock or vehicle control (DMSO) was added to the eluate to reach PROG final concentrations (0.004-40 µM), all dissolved within the same concentration of DMSO. Glass capillaries were prepared and loaded in triplicates into Prometheus (Nanotemper) and heated from 15° to 95 °C at a rate of 1°/min, the largest thermal scale available to run in the Nanotemper PR.therm-control software. The unfolding profile of a protein is a plot of its fluorescence signal against temperature. The protein is heated at a defined ramp over a specified temperature range. During the heating process, fluorescence is recorded at 330 and 350 nm. The resulting fluorescence ratio change is then analyzed for onset and inflection points. The temperature of onset and inflection is determined using built-in analysis function in Nanotemper PR.therm-control software. The onset of unfolding or aggregation is the temperature at which the channel conformation (unfolding or aggregation) process first starts. When comparing protein interactions with and without ligands, proteins can exhibit very similar Inflection points, yet their onsets of conformational changes can differ, or vise-vera, in this case, the delta between the two sets should be analyzed for binding evaluations. Onset and inflection points were validated by comparing the first-derivative variations between technical triplicates (high degrees of variations in the first derivative are shown in Supplementary Fig. 1b, c). CRC data obtained from microscale thermophoresis experiments were fitted to a Boltzmann function of the type:

$$y = \frac{A1 - A2}{1 + e^{x-x0}} + A2$$

using Origin 2020 software (OriginLab; OriginLab Corporation).

### Computational modeling and molecular dynamics
The $\beta_{1/4}$ structures and sequences were obtained from Alpha Fold P97678 and Q9ESK8.

Structures were visualized using MOE 2019.01 (Chemical Computing Group, Montreal, QC, Canada). Reliability of structure prediction in Alpha Fold was established by superposing $\beta_4$ structure from this resource on $\beta_4$ coordinates from cryo-electron microscopy[41] (PDB 6V22) and observing low root-mean-square deviation (RMSD) between two structures (Supplementary Fig. 2b). All docking, simulations, and scoring was done using MOE 2019.01. The structural library (MBD file) of 10 energetically minimized conformations (set to MMFF94X for small molecules, R field 1:80, cut off 8;10, set in an aqueous

environment) of PROG and PROG-oxime structures were used to dock these two steroids to all atoms and to transmembrane domains of the rat *KCNMB1* and *KCNMB4* proteins (AF-P97678-F1 Model, PDB P97678 and AF-Q9ESK8-F1 Model, PDB: Q9ESK8, respectively) using the Amber10:EHT force field (in a vacuum). The most energetically favorable conformations (RMSD < 2 Å, enclosed by the van der Waals radii +1.4 Å; ranked by lowest $S$ score, which represents the interaction strength, and the smallest E score (kcal/mol), which represents energy[48]) of PROG or PROG-oxime were used as starting points for the simulations at the theoretical binding sites, i.e., TMs, loop, with and without PROG or PROG-oxime pre-docked at the theoretical loop binding sites. The steroid structures were placed near the hypothesized docking regions (i.e., loop or TMDs) of the energetically minimized $\beta_1$ structure. Pre-docked structures (i.e., steroids within the loop) were energetically minimized as above. To detect specific amino acids that likely participate in ligand binding, all-atom simulations were run using Nosé-Poincaré-Andersen (NPA) equations of motion, under Amber10:EHT force field, R field 1:80, cutoff 8;10 with time steps of .001 ps, light bond constraints, sampling time of 0.1 ps, with checkpoints every 250 ps (STable 1). Charges were fixed, and no water molecules were present. Each run started with 10 ps-long sampling of the conformational space at 0 °K. After that, ligand-protein complexes were subjected to a 100 ps-long heating phase to 300 °K. Equilibration state lasted 100 ps (Supplementary Fig. 9), $T = 300$ °K. Production state was run at 300 °K, 100 kPa for 250 ps. This time is sufficient to capture positioning of the ligand within protein binding site[49]. For each type of steroid with respective protein, 3–7 independent simulation runs were performed.

### Electrophysiology data acquisition and analysis

SM cells were isolated from middle cerebral arteries (MCA) dissected out of 8–12 -week-old C57BL6 and *KCNMB1*[-/-] on C57BL/6J background male mouse as described[26]. Ionic currents were recorded from excised, inside-out (I/O) patches. I/O recordings were obtained at +30 mV to evoke robust (sub-maximal) channel activity from $\beta_1$ K/O SM[50]. Bath and electrode solutions contained (mM): 130 KCl, 5 EGTA, 1.6 HEDTA, 2.28 MgCl$_2$ (free [Mg$^{2+}$] = 1 mM), 15 HEPES; pH 7.40. In all experiments, free [Ca$^{2+}$] in solution was adjusted to the desired value by adding the appropriate amount of Ca$^{2+}$ from a 1 mM CaCl$_2$ stock. An agar bridge with Cl$^-$ as main anion was used as ground electrode. Solutions were applied onto the cytosolic side of I/O patches using an automated, pressurized system (Octaflow; ALA Scientific Instruments Inc.) through a micropipette tip with an internal diameter of 100 μm. Ionic currents at single channel resolution were recorded using an EPC8 amplifier (HEKA) at 1 kHz. Data were digitized at 5 kHz using a Digidata 1320A A/D converter and pCLAMP 8.0 (Molecular Devices); $n = 5$/group.

Patch-clamp and bilayer data were analyzed using Clampfit 10.7 software (Molecular Devices). NPo was used as an index of channel steady-state activity where $N$ = number of channels in the patch (defined as a maximal number of opening levels at Po≤1) and Po=single channel open probability. Drug-induced NPo changes were determined by comparing baseline NPo (i.e., NPo with the patch/cell perfused with bath solution) to the NPo from the same patch/cell exposed to the agent under study (DMSO/PROG/oxime) dissolved in bath solution. NPo and mean open time values were automatically calculated using Clampfit 10.7, the latter using a 50% threshold detection event.

### Cerebral artery diameter measurement in vitro

Eight-12 week-old C57BL/6J and *KCNMB1*[-/-] on C57BL/6 background mice of both sexes were deeply anesthetized with isoflurane via inhalation in a tightly sealed jar. Resistance size MCAs (~100 μm in external diameter) were dissected out of the mouse brains after animal euthanasia with sharp scissors. For de-endothelialized records, the endothelium was removed by passing an air bubble into the vessel lumen for 90 s[30]. This method has been consistently used by our group and

validated using endothelium-dependent versus endothelium-independent vasodilators[16,26]. Arteries were cannulated as previously described by our group[26,40]. MCAs were cut into 5 to 10 mm-long segments under a microscope (Nikon SMZ645). The segment was cannulated at each end, and the artery exterior was continuously perfused with physiologic sodium saline (PSS) of the following composition (mM): 119 NaCl, 4.7 KCl, 1.2 KH$_2$PO$_4$, 1.6 CaCl$_2$, 1.2 MgSO$_4$, 0.023 EDTA, 11 glucose, 24 NaHCO$_3$, pH=7.4. PSS was continuously bubbled with an O$_2$/CO$_2$/N$_2$ (21/5/74%) gas mixture and maintained at 35−37 °C. The artery external wall diameter was measured using the automatic edge-detection function of IonWizard software (IonOptix) via Sanyo VCB-3512T camera (Sanyo Electric Co.). Arteries were first incubated at an intravascular pressure of 10 mm Hg for 10 min. Then intravascular pressure was increased to 60 mmHg and held steady throughout the experiment to induce myogenic tone development and maintenance[19,30,51]. Each artery segment was exposed to PROG, PROG-oxime (in presence and absence of 1 μM paxilline) only once to avoid development of desensitization during repeated applications of PROG-containing solution. To determine arterial viability, arterial contractility was probed with a high KCl solution at the end of each experiment. The high KCl solution was composed of the following composition (mM): 63.7 NaCl, 60 KCl, 1.2 KH$_2$PO$_4$, 1.2 MgSO$_4$, 0.023 EDTA, 11 glucose, 24 NaHCO$_3$, 1.6 CaCl$_2$, pH=7.4. Like PSS, the high-KCl solution was continuously bubbled with O$_2$/CO$_2$/N$_2$ (21/5/74%) gas mixture and maintained at 35−37 °C.

### *KCNMB1*[-/-] mouse artery permeabilization

*KCNMB1*[-/-] ($\beta_1$-lacking) mouse MCAs were dissected out and permeabilized to allow delivery of pcDNA3 vector containing $\beta_1$ subunit-coding cDNA or cDNA coding mutated $\beta_1$W87F[20]. Permeabilization was performed following a standard chemical loading/reverse permeabilization technique[52,53].

### Cerebral artery diameter measurement through cranial window in vivo

Eight-12 week-old male and female C57BL/6J mice were anesthetized with a mixture of xylazine/ketamine (12/100 mg per kg) and kept anesthetized for the duration of the experiment with subsequent ketamine doses (50 mg/kg of weight) as needed. A catheter was inserted into the carotid artery so that the infusion went straight to the brain rather than towards the thoracic cavity. An area of the skull was cleared of tissue and thinned to expose the branching arteries originating from the MCA on the side the catheter was inserted, above the zygomatic arch, between the ear and eye of the skull[19,34]. The exposed arteries branching out from the MCA were monitored using a Leica MC170 HD microscope with a mounted camera (Leica M125 C) connected to a computer monitor. Drugs were diluted to their final concentration in sodium saline (0.9% NaCl) and administered via catheter at 0.1 ml/25 g of weight. Cranial window images before and after drug administration were acquired every 60 seconds for subsequent analysis.

### Chemicals

POPE, and POPS were purchased from Avanti Polar Lipids. All other chemicals were purchased from Sigma Aldrich. PROG or PROG-oxime was dissolved in DMSO and diluted into elution buffer, bath solution, *cis* chamber solution, PSS, or saline immediately before application to the protein, myocyte membrane, bilayer, arterial segment, or infused into mouse carotid artery, respectively. Each myocyte membrane, arterial segment, mouse, or artificial lipid bilayer was exposed to PROG or PROG-oxime only once to avoid possible desensitization.

### Data analysis and statistics

Statistical analysis was performed using InStat 3.05 (GraphPad). When the number of observations in the groups under comparison exceeded 6, and the Gaussian distribution of the data was confirmed by the

Kolmogorov–Smirnov test, analysis was performed using an unpaired Student's $t$ test. In all other cases, statistical analysis was conducted using the Mann–Whitney nonparametric test. For comparison of multiple experimental groups, the Kruskal–Wallis test with Dunn's post-test were used. In all cases, testing assumed two-tail $P$ values with significance set at $P < 0.05$. Validity of the fitting results were given by the corresponding reduced Chi-square and $R$-square values; the reduced Chi-square statistic was used to represent the goodness of fit testing[54] and the $R$-square results evaluated the scatter of data points around the fitted regression line[55].

### Reporting summary

Further information on research design is available in the Nature Portfolio Reporting Summary linked to this article.

## Data availability

All data that support the findings of this study are freely available from the authors upon request. The source data underlying Fig. 1b, c, e, f, h, i, k, l; 2e, f; 3d, e; 4a, e, f; 5c, d; 6c–g; 7c–f; and 8b, e, f and Supplementary Figs. 1a; 3a, b; 4a, b; and 6d are provided as a Source data file. Trajectory and molecule files for the computational modeling are available upon request. Source data are provided with this paper.

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

## Acknowledgements

The authors thank Sydney Hawks for her excellent technical assistance. This work was supported by the National Heart, Lung, and Blood Institute (Grant R01-HL147315; to A.M.D.).

## Author contributions

K.C.N.: designed research, performed research, analyzed data, wrote paper. A.C.S.: performed research, analyzed data. A.N.B and A.M.D.: designed research, contributed new reagents and analytic tools, wrote paper.

## Competing interests

The authors declare no competing interests.
