## [Peer Review File · Nature Communications]

Progesterone activation of β 1-containing BK channels involves two novel binding sitesReviewers' comments:

Reviewer #1 (Remarks to the Author):

The study describes the mechanism by which progesterone, a hormone that is widely used in hormone replacement therapy and in brain ischemic events, activates BK channels known to play a major role in vasodilation of cerebral arteries. The authors demonstrate that progesterone activates BK channels by binding with the regulatory subunits $\beta 1$ or $\beta 4$ and identify a putative binding site that is found in between the domains with progesterone acting as a bridge between them. This is interesting and potentially important. The main result that $\beta 1/\beta 4$ subunits are required is strong and the conclusion is well justified but there are also multiple concerns described below. The most major is lack of evidence that progesterone-BK binding plays a role in its therapeutic effects. There are also several other issues that need to be clarified with better presentation and additional data.

Specific Comments:

1. Experiments showing functional effect of progesterone on channel activity and the requirement for $\beta 1/\beta 4$ subunits are strong but the binding experiments using MST while interesting are presented with very scant details making it very difficult to judge whether the conclusions are justified. The raw traces shown in fig 2A seem to show an inflection in the presence of the hormone at a very high temperature, is it not unusual to go to such high temperatures? A trace for DMSO only seem to show no inflection – is this the justification that it indicates the binding event? The interpretation for dose response that has a biphasic phase is not clear (2B). It is mentioned that it is supposed to indicate two binding sites but this needs to be much better explained. It is also not clear why in 2C, it shows the average delta inflections in the presence of progesterone and DMSO respectively while the raw traces seem to show no inflection for DMSO at all. All this needs to be thoroughly clarified with additional control experiments that demonstrate the feasibility of the approach.
2. It would be really interesting to perform the simulations in temperatures that match the MST experiments and demonstrate that the binding events are predicted to match the MST results. That would be a strong confirmation of these data.
3. "at low PROG concentrations, the double-mutant, $\beta 1Y32V+W163I$, was still destabilized by PROG binding (Fig. 2D; $P=0.031$)" – I do not see these data in Figure 2D.
4. A part of the study that proposes the existence of a high-affinity site lacks any functional data without which, the functional significance is not clear.
5. Functional therapeutic effect of progesterone-BK binding should be demonstrated in the appropriate animal model
6. The presentation and the logical flow should be significantly improved. Specifically:
 - a. While the authors clearly show that progesterone activates BK in $\beta 1/\beta 4$ -dependent manner, the presentation of the data is not logical. To improve clarity, experiments that demonstrate the essential role of $\beta 1/\beta 4$ subunits in progesterone activation of BK, currently figure 1 and figure 2G-I should be shown together, and a full dose response should be presented. Currently, a partial dose response comparing 10 μM and for 0.01 μM is shown in figure 2D. It is also not clear what is the meaning of the term "embolden", what the data show is that $\beta 1$ or $\beta 4$ subunits are required for BK to respond to the hormone.
 - b. More details are also needed for the computational studies. From the methods section, it appears that it was done using docking followed by atomistic simulations using Amber force field. This is a strong approach. However, it is not clear and needs to be better defined what is the energetics of the low vs. high affinity sites. In terms of the clarity of the presentation, the docking poses, currently presented in figure 2e and f, should be shown together with the mutants that test the predictions of

the model currently shown in figure 3A-D.

Reviewer #2 (Remarks to the Author):

This manuscript by North et al shows an interesting result that PROG modulation of BK channels depending on BK channel regulatory beta subunits. Microscale thermophoresis, molecular modeling and docking, mutagenesis and electrophysiology methods were used to identify two binding sites, one of which is "high affinity" and the other is "low affinity". The "high affinity" binding is "permissive" such that the binding of PROG to this site alters the beta 1 conformation that allows the binding of PROG at the "low affinity" site, which modulates channel opening.

The binding sites of PROG on the beta subunits were identified by docking on computation model of the structures, which need experimental validation. Nevertheless, the experimental validation of these results is limited to a few mutational studies. Particularly, experimental evidence is lacking for the major conclusion on the "high affinity PROG binding" and its "permissive role". Some important control in experiments are missing, and some data do not provide clear-cut conclusions (see detailed comments).

1. Comments on methods.

A) It is not clear how beta subunit structures were obtained. Detailed methods need to be described. Important parameters and restraints need to be described such that the models can be evaluated and reproduced.

B) The "low affinity" site located in the membrane spanning helices. The interaction between PROG and beta subunits is thus influenced by membrane lipids. In addition, when beta subunits associate with Slo1 there may be interactions between Slo1 and beta subunits at the binding site. Docking of PROG on this site is not reliable without considering lipid effects or Slo1 interactions.

C) Likewise, microscale thermophoresis measurements of PROG binding is based on purified proteins, and the effects of lipid membrane or Slo1 on the experimental results are not assessed. By the way, the data to show the quality of purified proteins are missing.

D) The physical basis of microscale thermophoresis measurements of PROG binding is not well explained. In Fig 2C and 4C, why does the "delta range" indicate PROG binding? The authors suggest two PROG binding sites based on Fig 2B. Presumably the data in Fig 2C and 4C were derived from the data in Fig 2B, but can the authors plot the results for the two binding sites separately in Fig 2C and Fig 4C?

2. Page 7 bottom, " β 1Y32V+W163I, was still destabilized by PROG binding (Fig. 2D; $P=0.031$)."

Fig 2D does not show this. In addition, the legend for Fig 2D is confusing, which does not match the figure.

3. No experimental data are shown to support the role of the "high affinity site" in channel function or its "permissive role" for PROG modulation of channel function. The only evidence cited in the text is Fig 2D, but no data in Fig 2D matches the description. Fig 4C shows some data that may match the description in the text, but these data alone may not provide any clear-cut conclusion. In addition, if the high affinity binding is so important for PROG modulation of the channel as Fig 4 simulations suggested, a mutation at the putative binding site should abolish the effects of PROG on Slo1+beta1 channel opening. This should be an easy experiment to validate the mechanism.

4. Do Beta 1 Y32V and W163I still associate with Slo1 and modulate the channel? This needs to be verified.

5. Fig 3C: in this experiment a positive control of PROG on the WT beta1 performed in the same batch of experiment would be important.

6. Fig 3D: For the mutant beta 1 subunits are the data shown with PROG? If not the data with PROG need to be shown to compare with the data for the WT beta 1.

7. In Fig 1A please indicate the meaning of "I" and "O"

Reply to reviewers' comments:

Reviewer #1: We very much thank the reviewer for underscoring that (a) our demonstration of progesterone activation of BK channels by binding with their regulatory β_1 or β_4 subunits and the identification of a novel binding site in between the two transmembrane domains of these subunits with progesterone acting as a bridge between them “*is interesting and potentially important*”; (b) the main result showing that β_1/β_4 subunits are required for progesterone action “*is strong*” and (c) “*the conclusion is well justified.*”

Many new experiments comprising probing aldosterone vasodilatory action at the organ level *in vitro* (isolated brain arteries), *in vivo* (brain circulation evaluation through a cranial window), and potential therapeutic use of progesterone in a pharmacological model of brain cerebrovascular constriction were included to address the “*lack of evidence that progesterone-BK binding plays a role in its therapeutic effects.*” This and other specific concerns have been addressed in their entirety as described below.

Specific Comments:

1. “...*the binding experiments using MST while interesting are presented with very scant details making it very difficult to judge whether the conclusions are justified. The raw traces shown in fig 2A seem to show an inflection in the presence of the hormone at a very high temperature, is it not unusual to go to such high temperatures?*”

We thank the reviewer for the suggestion on how to add clarity and have now added details regarding the MST studies in the Materials and Methods section (p. 18 last 2 lines from bottom to p. 19, 1st parag.). Indeed, temperature of inflection is fairly high. However, it reflects detectable the net changes in the protein population. In addition, individual proteins may undergo conformation transitions upon ligand binding at lower temperatures. More importantly, MST does not measure the temperatures at which a binding event occurs, but rather characterizes how protein sensitivity to temperature-driven unfolding is modified when the ligand is already bound.

“*A trace for DMSO only seem to show no inflection –is this the justification that it indicates the binding event?*”

We thank the reviewer for the suggestion on how to add clarity and thus we have added details in the Materials and Methods section regarding the MST studies (p. 18 to p. 19). Last but not least, we have replace the DMSO trace in question to a more representative trace (new **Fig.1A**). Thank you for the observation.

“*The interpretation for dose response that has a biphasic phase is not clear (2B). It is mentioned that it is supposed to indicate two binding sites but this needs to be much better explained.*”

As requested, we now provide a more detailed description of the biphasic nature of the binding curve, and its interpretation as two distinct binding processes (p. 5, l. 5 from bottom to p. 6, l. 1-2). Our support for the two PROG binding sites is highlighted in **new Fig.4A**; the double-mutant (Y32V_W163I), which disrupts the low-affinity site, still binds low progesterone concentrations.

“*It is also not clear why in 2C, it shows the average delta inflections in the presence of progesterone and DMSO respectively while the raw traces seem to show no inflection for DMSO at all. All this needs to be thoroughly clarified with additional control experiments that demonstrate the feasibility of the approach.*”

Fig. 1A right panel shows that there is inflexion with DMSO. However, additional control experiments have been added in a new figure (**Suppl. Fig. 1**) demonstrating the behavior of the

system when boiled protein subunit of interest or solution with no protein are investigated. Under these two conditions, data drastically differ from those with wt β_1 protein \pm progesterone or DMSO (**Fig. 1A**). The onset and inflection point are not sensed or recorded by the program at all when no protein is present.

2. *“It would be really interesting to perform the simulations in temperatures that match the MST experiments and demonstrate that the binding events are predicted to match the MST results. That would be a strong confirmation of these data.”*

MST does not reflect that a binding event(s) occurs at a specific temperature; rather, it shows how the thermal sensitivity of the protein changes when a binding event already occurred. If binding does not occur, then the thermal unfolding of the protein would match that of the protein without the ligand. What the reviewer is suggesting could eventually be done with molecular dynamics simulations: we could dock progesterone on β_1 subunits, run MD at room or physiological temperature, and then heat up this molecular complex to temperatures that are used during MST to determine whether the protein structure collapses at the same temperature as observed during MST. If this happened, we failed to see what this information would add anything substantive to what we already established. More importantly, both MST and MD have their limitations: for MST, proteins are isolated, without the complex proteolipid environment that exists in their native membranes; for MD, although the absence of a complex environment could be overcome up to a certain degree, the appropriateness of force-field descriptors becomes a major limitation that increases as molecular complexity does. Therefore, these methodological limitations will make it more likely that MST and computational results will not match. For these reasons, we preferred to validate our MST and/or computational results with mutagenesis and functional studies. The validation of MST data with negative controls (no protein and boiled protein) are shown in **new SFig.1**. In addition, we have now added more electrophysiological data (**new Fig.4D-F**) and experimental results from arteries *ex vivo* (**new Fig.6G**) to validate our basic conclusions on the high-affinity site obtained the initial MST approach.

3. *“at low PROG concentrations, the double-mutant, $\beta_1Y32V+W163I$, was still destabilized by PROG binding (Fig. 2D; $P=0.031$)” – I do not see these data in Figure 2D.”*

We apologize for the typo; we referred to previous **Fig. 4C** (current **Fig. 4A**).

4. *“A part of the study that proposes the existence of a high-affinity site lacks any functional data without which, the functional significance is not clear.”*

New experiments were performed to construct a concentration-response curve for progesterone-induced modification of $\alpha+\beta_1$ channel activity. Results show that progesterone at the low concentrations that binds the β_1 high-affinity site (**Fig. 1B,F**) does not increase activity of these heteromeric channels (**new Fig. 1B**). In addition, new experiments were conducted to probe progesterone on a construct where a key residue in the high-affinity binding site was substituted (β_1W87F). Data clearly show that this point mutation blunts the activation of the resulting heteromeric channels by 10 μ M progesterone, i.e., steroid concentrations that bind the low affinity site (**new Fig. 4D,E,F**). We further validated these finding by comparing the effects of 10 μ M PROG in pressurized, middle cerebral arteries (MCA) isolated from *KCNMB1*^{-/-} mice, permeabilized with cDNA encoding wt β_1 and *KCNMB1*^{-/-} MCAs permeabilized with cDNA encoding β_1W87F mutant (**new Fig. 6G**). Remarkably, *KCNMB1*^{-/-} arteries permeabilized with β_1W87F cDNA failed to respond to 10 μ M PROG, while *KCNMB1*^{-/-} arteries permeabilized with wt β_1 significantly dilated. This dilation was no different from the diameter responses of de-endothelialized, pressurized MCA from wt (C57BL/6J) mice (**new Fig. 6A-E**). Collectively all these new results indicate that the high-affinity binding site located in the beta subunit loop is “permissive” so that the binding of progesterone to this site allows the binding of this steroid at the low affinity site, which modulates channel gating. Computational molecular dynamics

simulations show that upon progesterone binding to the high-affinity site in the β_1 subunit loop, transmembrane domains of this protein come closer to each other. Hence, progesterone molecule can now bridge both transmembrane domains (low-affinity site) and influence channel activity. The text has been edited accordingly to incorporate all these new data and make our conclusion on this point clearer (rewritten pages 9 and 10, and page 12, 2nd Parag.).

5. “*Functional therapeutic effect of progesterone-BK binding should be demonstrated in the appropriate animal model.*”

We addressed this concern by conducting three different types of experiments. Our model rests its experimental validation on the differential efficacy of progesterone vs. its oxime on β_1 -containing BK channels: increased activity and no effect respectively (current **Fig.3**). BK channel activation in arterial smooth muscle (SM) leads to endothelium-independent MCA dilation (reviewed in *Dopico et al., 2018*). Thus, our new experiments demonstrate that progesterone but not its oxime evoke reversible dilation of (1) *in vitro*-pressurized, de-endothelialized, MCA isolated from mice (**new Fig.6**), a system where arterial dilation is primarily determined by the activity of SM (β_1 -containing) BK channels (reviewed in *Jaggar et al., 1998; Dopico et al., 2018*), and (2) pial arteries branching from MCA evaluated *in vivo* using a cranial window approach (**new Fig.7**). Lastly, (3) we used a pharmacological tool to induce MCA constriction: acute administration of alcohol at concentrations found in circulation during moderate-heavy episodic drinking (i.e., 50 mM ethanol) evokes a reversible constriction of all branches of the Willis’ circle whether evaluated in mouse or rat (*Mysiewicz et al., 2023*), an ethanol action that is mediated by the BK channel β_1 TM2 (*Bukiya et al., 2009; Kuntamallappanavar et al., 2017*), and can be reversed by BK β_1 -targeting agents (*North et al., 2020*). The new data demonstrate that alcohol-induced MCA constriction is, indeed, blunted at all concentrations (10-75 mM ethanol) by 10 μ M progesterone, a concentration that activates β_1 -containing BK *via* interaction with the low affinity site (new **Fig. 8A-B**). Moreover, this effect was replicated *in vivo* (new **Fig. 8C-F**).

6. “*The presentation and the logical flow should be significantly improved. Specifically:*

a. *While the authors clearly show that progesterone activates BK in β_1/β_4 -dependent manner, the presentation of the data is not logical. To improve clarity, experiments that demonstrate the essential role of β_1/β_4 subunits in progesterone activation of BK, currently figure 1 and figure 2G-I should be shown together, and a full dose response should be presented. Currently, a partial dose response comparing 10 μ M and for 0.01 μ M is shown in figure 2D. It is also not clear what is the meaning of the term “embolden”, what the data show is that β_1 or β_4 subunits are required for BK to respond to the hormone.”*

We conducted new experiments to comply with the request for establishing a full CRC to the effect of progesterone on heteromeric β_1 -cbv1 channels; data show that, indeed, the steroid significantly increases channel activity at concentrations that bind the low affinity site identified in this subunit (i.e., tens of μ M) (new **Fig. 1**).

We thank the reviewer for the suggestion on how to improve the logical flow of our data presentation. Thus, we start with **Fig. 1** combining MST and corresponding functional validation using reconstituted channels that contain β_1 or β_4 in lipid bilayers (vs. cbv1 alone); **Fig. 2** follows with the role of the low-affinity site that is conserved in these two subunits; thus **Fig. 3** validates the resulting “bridge model” from Fig. 2; **Fig. 4** brings on the importance of the high affinity site; **Fig. 5** validates the previous data in a more integrative system: natural MCA SM membranes expressing native BK channels; we end with **Figs. 6-8** which further validate our previous molecular/isolated membrane findings into *in vitro* and *in vivo* models of MCA (arterial) function and the vessel response to a cerebrovascular constrictor. We deeply thank the reviewer for

noting this deficit of logical presentation in the previous version. Following the logical re-ordering the paragraph including the term “embolden” was deleted.

b. *“More details are also needed for the computational studies. From the methods section, it appears that it was done using docking followed by atomistic simulations using Amber force field. This is a strong approach. However, it is not clear and needs to be better defined what is the energetics of the low vs. high affinity sites.”*

We thank the reviewer for the suggestion on how to add clarity and have added details in the Methods section for the computational and molecular dynamics studies (p. 18, lines 1-2 from bottom to page 19, 1st Parag., and p. 19 last Parag. to p. 20, 1st Parag.).

“In terms of the clarity of the presentation, the docking poses, currently presented in figure 2e and f, should be shown together with the mutants that test the predictions of the model currently shown in figure 3A-D.”

We thank the reviewer for this suggestion. The figures in question have been rearranged following the reviewer’s indication resulting in the new **Fig. 2**.

Reviewer #2:

We deeply thank for the reviewer for highlighting the multi-methodological approach used in our study to show that “*PROG modulation of BK channels depends on BK channel regulatory beta subunits*” and “*to identify two binding sites, one of which is “high affinity” and the other is “low affinity”*”. The “*high affinity*” binding is “*permissive*” such that the binding of PROG to this site alters the beta 1 conformation that allows the binding of PROG at the “*low affinity*” site, which modulates channel opening”, and finding this study “*interesting*”.

We conducted new experiments to address the two main concerns of the reviewer, i.e., “*experimental evidence is lacking for the major conclusion on the “high affinity PROG binding” and its “permissive role”*”...“*some important control in experiments are missing*”, and “*some data do not provide clear-cut conclusions*”, as detailed below in our reply to the specific comments.

1. Comments on methods.

A) “*It is not clear how beta subunit structures were obtained. Detailed methods need to be described. Important parameters and restraints need to be described such that the models can be evaluated and reproduced.*”

The origin of the β subunits has been given in the text (p. 17, 3rd Parag., l. 1-5, and p. 19, 2nd Parag., l. 1-3). We now add clarity and details in the Material and Methods section for the computational docking and molecular dynamics studies (p. 19, last Parag., to p. 20, 1st Parag.).

B & C) “*The “low affinity” site located in the membrane spanning helices. The interaction between PROG and beta subunits is thus influenced by membrane lipids. In addition, when beta subunits associate with Slo1 there may be interactions between Slo1 and beta subunits at the binding site. Docking of PROG on this site is not reliable without considering lipid effects or Slo1 interactions*” and “*Likewise, microscale thermophoresis measurements of PROG binding is based on purified proteins, and the effects of lipid membrane or Slo1 on the experimental results are not assessed. By the way, the data to show the quality of purified proteins are missing.*”

The reviewer is absolutely right in underscoring a putative role of lipids in slo1- β_1 interactions and heteromeric channel-progesterone interaction. Regarding the latter, we need to underscore that progesterone action on these heteromers is similar in native SM membranes (new **Fig. 5**) and artificial POPE/POPS planar bilayer (new **Fig. 1**), i.e., two systems of different lipid composition. Moreover, our results indicating activation by progesterone through the β_4 subunit in artificial POPE/POPS planar bilayers validate King’s findings in oocytes (*King et al., 2006*), which again reflect conservation of the basic phenomenology in two completely different lipid environments. Thus, the effects of PROG are neither mediated by disruption of the lipid environment of the channel complex nor drastically altered by such environment. Regarding the former, it is noteworthy that slo1(cbv1)- β_1 functional coupling still occurs in a binary bilayer (new **Suppl. Fig. 3**), i.e., of lipid composition different from that of arterial SM where BK channels regulate SM function. Therefore, changes in lipid composition, while potentially important, do not affect the basic question under study: direct, functional interaction between progesterone and the BK channel subunits that define the SM channel phenotype. Regarding the computational docking of progesterone and microscale thermophoresis data, we need to underscore that, lipids can be included neither in our computational docking/simulation nor in microscale thermophoresis assays. However, docking of progesterone was conducted in vacuum and hydrophobic media, showing identical poses in both media. Moreover, if we were to profile the effects on lipid in MST, we would have >1K lipid/types to profile. This would be a study on its own and is unfeasible for the current study to include.

Last but not least, we conducted new MST studies, which show the behavior of the system in absence of β_1 and when β_1 is pre-boiled before the assay. Results clearly show the behavior of these two systems differ from that of unboiled, folded/unfolded β_1 (new **Suppl. Fig 1**)

D) *“The physical basis of microscale thermophoresis measurements of PROG binding is not well explained. In Fig 2C and 4C, why does the “delta range” indicate PROG binding? The authors suggest two PROG binding sites based on Fig 2B. Presumably the data in Fig 2C and 4C were derived from the data in Fig 2B, but can the authors plot the results for the two binding sites separately in Fig 2C and Fig 4C?”*

MST is a rapidly emerging and powerful technique which is optimal for detecting and quantifying biomolecule interactions by the thermophoretic detection of time-dependent changes in protein intrinsic fluorescence related to conformation, charge, and size of a molecule as they are induced by a binding event. The unfolding profile of a protein comes from a plot of protein-associated fluorescence against temperature. To get this plot, the protein is heated at a defined ramp over a specified temperature range. During the heating process, fluorescence is recorded at 330 nm, which represents the fluorescence of burred aromatic amino acids, and at 350 nm, which represents thermal unfolding and exposure of aromatic amino acids. The resulting fluorescence ratio change is then analyzed for onset and inflection points. The temperature of onset and inflection is determined using built-in analysis function in Nanotemper PR.therm-control software. The onset of unfolding or aggregation is the temperature at which the protein conformation (unfolding or aggregation) process first starts while the inflection point determines when the speed of conformational changes is at its greatest. When comparing protein interactions with and without ligands, proteins can exhibit very similar onset and inflection. Thus, when examining ligand-protein interactions, as in the present study, the delta parameter (as defined in Materials and Methods) provides the most accurate indicator of ligand binding to the protein (Duhr & Braun, 2006; Wienken et al., 2010; Jerabek-Willemsen et al., 2011; Seidel et al., 2012; 2013; Zillner et al., 2012). A concise version of this explanation has been added to Materials and Methods (p. 18 to p. 19). Additionally, our support for the two PROG binding sites is highlighted in **new Fig.4A**; the double-mutant (Y32V_W163I) which affects the low-affinity sites, still binds at low concentrations. Lastly, we prefer not to plot the binding events for the high- and low-affinity sites separately, yet we now further elaborate on the interpretation of the biphasic plot (previous **Fig. 2B**), as requested by Rev.#1 (Specific comment 1, 3rd parag.).

2 & 3. *“Page 7 bottom, ‘ β_1 Y32V+W163I, was still destabilized by PROG binding (Fig. 2D; $P=0.031$).’ Fig 2D does not show this. In addition, the legend for Fig 2D is confusing, which does not match the figure” and “No experimental data are shown to support the role of the ‘high affinity site’ in channel function or its ‘permissive role’ for PROG modulation of channel function. The only evidence cited in the text is Fig 2D, but no data in Fig 2D matches the description. Fig 4C shows some data that may match the description in the text, but these data alone may not provide any clear-cut conclusion. In addition, if the high affinity binding is so important for PROG modulation of the channel as Fig 4 simulations suggested, a mutation at the putative binding site should abolish the effects of PROG on Slo1+beta1 channel opening. This should be an easy experiment to validate the mechanism.”*

We thank the reviewer for noting this error: we referred to previous **Fig. 4C** not **2D**. We fixed this typo and also corrected the respective legend (current **Fig. 4**). We also conducted the experiment proposed by the reviewer to document experimentally the permissive role of the high-affinity site in aldosterone action on the low affinity site. The new data demonstrate that the W to F substitution in the β_1 high-affinity site, while still sustaining functional coupling between this regulatory subunit and cbv1 (new **Suppl. Fig. 3**) blunts activation of the resulting cbv1+ β_1 heteromers by progesterone concentrations that bind to the low-affinity site (new **Fig. 4**). We further validated these finding by comparing the effects of 10 μ M PROG in pressurized,

middle cerebral arteries (MCA) isolated from *KCNMB1*^{-/-} mice, permeabilized with cDNA encoding wt β_1 and *KCNMB1*^{-/-} MCAs permeabilized with cDNA of β_1 W87F mutant construct (new **Fig. 6G**). Remarkably, *KCNMB1*^{-/-} arteries permeabilized with β_1 W87F cDNA failed to respond to 10 μ M PROG, while *KCNMB1*^{-/-} arteries permeabilized with wt β_1 cDNA significantly dilated. This dilation was not different from the arterial responses of de-endothelialized, pressurized MCA from wt (C57BL/6J) mice (new **Fig. 6A-E**). We deeply thank the reviewer for suggesting this experimental validation of our model.

4. “Do Beta 1 Y32V and W163I still associate with Slo1 and modulate the channel? This needs to be verified.”

New data shown in **Suppl. Fig. 3** document that the functional phenotype of cbv1+these mutated β_1 s can only result from effective cbv1- β_1 coupling: at constant voltage and internal calcium, (a) heteromeric steady-state activity is larger than that of homomeric channels and, more importantly, (b) mean open time from heteromers is significantly longer than that of homomers, consistent with all previous literature using cbv1 or other slo1 channel-forming subunits (e.g., Fig. 2 in *Bukiya et al., 2009*).

5. “Fig 3C: in this experiment a positive control of PROG on the WT beta1 performed in the same batch of experiment would be important.”

The requested data have been added (see new **Fig. 4**).

6. “Fig 3D: For the mutant beta 1 subunits are the data shown with PROG? If not the data with PROG need to be shown to compare with the data for the WT beta1.”

Thank you for noting the omission: data with the mutants were obtained in presence of progesterone and thus, the labels to the x-axis of the plot has been accordingly modified; in addition, we have added the results from the mutants in presence of DMSO (new **Fig. 2F**).

7. “In Fig 1A please indicate the meaning of ‘I’ and ‘O’.”

The meaning of ‘I/O’ has now been added to the main text (p. 11, l. 4) and the caption (p. 35, l. 9 from bottom).

REVIEWER COMMENTS

Reviewer #1 (Remarks to the Author):

The revised manuscript is significantly improved but a major concern about the role of progesterone binding to BK channels in its therapeutic effects is only partially addressed.

The new data show that while progesterone induces vasodilation of cerebral artery, its structural analogue does not. This correlates with a lack of the analogue binding to BK channels. This is a promising result but there could be many more differences between progesterone and its analogue in interacting with various targets and thus this approach does not provide strong evidence for the involvement of BK channels. A decisive answer may come from identifying mutations in the channels that interfere with progesterone binding and generating a model to test whether a therapeutic effect of progesterone is lost.

Reviewer #2 (Remarks to the Author):

Additional experiments and clarifications by the authors are appreciated. However, concerns on the methods and on the high affinity PROG site remain. The following are additional comments.

Responses to comments on methods:

B&C: The authors pointed out that lipid species do not alter experimental results, but these results do not imply that lipids are not important in simulations. The concerns remain that the simulations without including lipids may introduce errors. The same concern on MTS also remains. Some additional validations may help (see below).

sFig 1A: more data points are needed for the boiled beta1 to support the conclusion.

Fig 4A: in 4.1 μM PROG the double mutation did not reduce the temperature range but did in 0.004 μM . Should the high affinity site be saturated in 4.1 μM PROG as well, but why did the binding of the high affinity site have no effect?

Fig 4B: The structures of the beta subunits in this manuscript were predicted using alpha-fold, which need to be validated. Since the cryo-EM beta4 structure was published already (<https://doi.org/10.7554/eLife.51409>) the authors need to compare their model with the published structure and validate both the extracellular and membrane-spanning domains, where the high and low affinity PROG binding sites are located.

Fig 4B: In the cryo-EM structure the extracellular domains of the neighboring beta4 subunits form an interface. The authors may want to indicate whether and how the docking of PROG is affected by the interface.

Fig 4A: in the low affinity site W32F seemed not to disrupt PROG binding, but in the high affinity site W87F acted differently. The authors may want to perform the docking simulations of both W37F and W87F to match with the experimental observations. This would be a demonstration of the validity of the docking approach.

Reply to Reviewer #1.

“The new data show that while progesterone induces vasodilation of cerebral artery, its structural analogue does not. This correlates with a lack of the analogue binding to BK channels. This is a promising result but ...does not provide strong evidence for the involvement of BK channels. A decisive answer may come from identifying mutations in the channels that interfere with progesterone binding and generating a model to test whether a therapeutic effect of progesterone is lost.”

We concur with the reviewer on the key importance of demonstrating that relevant mutations alter a functional effect *“of progesterone in the appropriate animal model.”* Therefore, in this re-revision, we electroporated middle cerebral arteries from beta1 subunit knock-out mice with pcDNA3.2 vectors carrying cDNA encoding single-point mutants shown already in the previous two versions of our study to blunt both binding to and activation of BK channels by PROG (Y32V and W163I, these mutants belonging to low-affinity binding site, which is the effector of progesterone action on BK channels). The new data show that *in vitro* pressurized arteries containing either point substitution, in contrast to those containing wt beta1 subunits, fail to dilate in response to progesterone! In contrast, cerebral artery constriction by ethanol is not influenced by these mutations (new panel **G** in **Fig.6** and new **SFig.7**). Constriction of cerebral arteries by 50 mM ethanol is widely known to result in brain ischemia in several species, including humans (e.g., Altura & Altura, 1982; Volkow et al., 1988; Liu et al., 2004; Sullivan et al., 2021; Mysiewicz et al., 2023), this alcohol action requiring beta1 subunit TM2, albeit a bridging model is not possible for a short 2 C molecule such as ethanol (Bukiya et al., 2009; Kuntamallappanavar & Dopico, 2017). These data, in conjunction with results presented with the previous version documenting the efficacy of progesterone and inefficacy of its close structural analog progesterone-oxime to respectively counteract cerebral ischemia by 50 mM ethanol both *in vivo* (**Fig.7**) and *in vitro* (**Fig.8**), clearly make the point on the importance of progesterone-BK binding via our bridge model to a potentially therapeutic use of this steroid (counteract cerebral artery constriction induced by toxicological/other means). Lastly, in line with the Editors' suggestion, we have “toned down” our claims on the role of progesterone-BK binding in eventual therapeutic effects of this steroid (p. 3, l. 1-2 from bottom of paragraph; Discussion, last 4 lines from bottom of section).

Reply to Reviewer #2.

1-“The authors pointed out that lipid species do not alter experimental results, but these results do not imply that lipids are not important in simulations. The concerns remain that the simulations without including lipids may introduce errors.”

The reviewer is absolutely right in raising the possibility that computational simulations of membrane spanning proteins in absence of lipids might introduce errors. Several factors, however, should be taken into account. First, a major experimental challenge arises from the plethora of lipid species present in all natural, including vascular smooth muscle, membranes. For example, “*The variation in headgroups and aliphatic chains allows the existence of >1,000 different lipid species in any eukaryotic cell*” (van Meer et al., *Nat Rev Mol Cell Biol.* 2008:9:112-124). With such diversity and lack of documented (peer-reviewed articles or otherwise) data on the specific lipid surroundings of beta1 subunits in native cerebral artery myocytes, it is impossible to guess what is the correct lipid composition to include in our simulations. Moreover, we cannot rely on data from transmembrane proteins other than beta1 because each protein (sub)family is believed to have its unique “lipid fingerprint” by sorting its lipid species within surrounding membrane domains (Corradi et al., *ACS Cent Sci.* 2018:4:709–717). Second, there are additional shortcomings in computational approaches when dealing with modeling proteins within a lipid bilayer-proteolipid environment. These include limited knowledge on the lipid composition of each membrane leaflet, i.e., existence of lipid horizontal domains, lack of agreement on the best approach for protein placement into the model membrane, and the use of different force-fields which best describe protein behavior vs. lipid media. Third, we cannot ignore the high computational cost for fine-grain simulations (Muller et al., *Chem Rev* 2019:119:6086–6161). Last but not least, there is a differential location of distinct steroid species in membranes of different lipid composition showing that even tiny differences in substituents on the steroid core result in significant changes in steroid-membrane interactions (Crowley et al., 2022), yet there are not available data for progesterone in smooth muscle membranes. Given the aforementioned uncertainties, dilemmas on experimental choices, and technical challenges, addressing the role of lipids on progesterone-beta1 docking would require, even if the major gaps in knowledge indicated above were solved, an entirely separate study (e.g., Sonntag et al., *Nat Commun.* 2011:2:304; Tieleman et al., *Biophys Rev.* 2021:13:1019-1027). In conclusion, we agree with the reviewer on the importance of membrane lipids. However, a systematic study on the influence of membrane lipids on progesterone-BK beta1 direct interactions is beyond the scope of our ground-breaking study which documents such interaction for the first time.

2-“The same concern (as above) on MTS also remains. Some additional validations may help (see below).”

Indeed, we have considered adapting microscale thermophoresis measurements to proteins incorporated into lipid spheres. However, there are several major technical limitations with this approach. The loading of beta1 protein-containing lipid spheres (liposomes) into the thermophoresis capillaries will result in: a) high background noise; b) loss of protein-associated signal due to lipid shielding, and 3) interference with protein unfolding from lipid melt. These limitations will simply preclude data acquisition. However, we thoroughly followed with additional validations requested by reviewer, as detailed below.

3-“SFig 1A: more data points are needed for the boiled beta1 to support the conclusion”.

We apologize for the paucity of initial data. We now performed additional experiments and increased the number of observations for thermal unfolding of boiled beta1 samples. Please see updated **SFig.1A**. Even with this larger data sample, “boiled” beta1 data displayed a much larger dispersion than nonboiled beta1, as previously shown with the smaller data set. This outcome is consistent with the high conformational variability of “boiled” protein preparations (i.e., subjected to

a heating/cooling cycle before loading into thermophoresis capillaries). Thus, this increased data variability is the expected result of heating/cooling, as a fraction of beta1 proteins can quickly refold in cell-free environment. To better illustrate the variability of boiled beta1 data we now present additional original trace of thermal unfolding for boiled beta1 (**SFig.1B**). Now, original traces in this figure show two examples of traces corresponding to an extreme versus a close-to-the average datapoint in the distribution (**SFig.1B**).

4-“Fig 4A: in 4.1 μM PROG the double mutation did not reduce the temperature range but did in 0.004 μM . Should the high affinity site be saturated in 4.1 μM PROG as well, but why did the binding of the high affinity site have no effect?”

The double mutation, as expected, does not affect the binding of 0.004 μM PROG, as the double substitution under consideration belongs to the low affinity site. It is puzzling to us, as surely is for the reviewer, that the double mutation in the low affinity site prevents 4.1 μM PROG from binding because, as the reviewer notes, the high affinity site should be saturated by PROG. We can only speculate: a possibility is that high levels of PROG introduces a shift of BK channels to some form of desensitized state preventing further binding (this fact was reported by the Magleby group demonstrating that high levels of intracellular calcium shifts BK channels to a desensitized, low affinity state(s); discussed in *Dopico and Lovinger, 2009*, and applicable to beta1-mediated, alcohol modulation of BK channels; *Kuntamallappanavar & Dopico, 2016*). A second speculation is that the binding of high levels of PROG to the mutated beta1 introduces another site for PROG binding to this mutant, yet to be identified. Interaction of PROG with this additional site may provide an effect on thermal unfolding behavior that is opposite that provided by PROG interaction with its high affinity site in the loop. Thus, we foresee a “zero sum” effect on ΔT with the double mutant at high concentrations of PROG. We prefer not to elaborate on these possibilities in the main text due to their highly speculative nature, unless the reviewer wants us to do so.

5-“Fig 4B: The structures of the beta subunits in this manuscript were predicted using alpha-fold, which need to be validated. Since the cryo-EM beta4 structure was published already (<https://doi.org/10.7554/eLife.51409>) the authors need to compare their model with the published structure and validate both the extracellular and membrane-spanning domains, where the high and low affinity PROG binding sites are located.”

We thank the reviewer for noting this. Since there is a beta4 subunit structure available from a cryo-EM study (*Tao & MacKinnon, 2019*), we have now superposed beta4 protein fold generated by Alpha Fold on beta4 structure uploaded from Protein Data Bank (ID 6V22). The superposition results are now presented in a new figure (**SFig.2A**; please note that due to the addition of this new SFig, numbering of the remaining SFIGs has been shifted). **SFig.2A** clearly demonstrates that the cryo-EM folding of beta4 and the folding predicted by Alpha Fold are in great agreement. Indeed, root-mean-square deviation estimates for these structures are low, as evident from the chromatic scale (green color corresponding to low root-mean-square-deviation values in the figure). There is only one point of rising discrepancy (in the loop area), due to the variability in single amino acid between the beta1 clone used in our work and the clone from cryo-EM publication referenced by the reviewer. Thus, based on beta4 observations, Alpha Fold provides an accurate prediction of protein folding. These new data are in line with a recent comprehensive assessment of Alpha Fold performance, which found Alpha Fold accuracy being “competitive with experimental structures in a majority of cases and greatly outperforming other methods” (*Jumper et al., Nature, 2021:596:583–589*). In this assessment, Alpha Fold structures had a median backbone accuracy of 0.96 Å root-mean-square deviation at 95% residue coverage. With this assurance in hand, we superposed beta1 structure predicted by Alpha Fold with the beta4 structure that was also predicted by Alpha Fold (new **SFig.2B**). The chromatic coding in this figure makes it clear that the transmembrane areas of these proteins are those that superposed mostly with very low root-mean-square deviation, owing to the high degree of amino acid similarity between two proteins; 44% of amino acid similarity

has been reported between beta1 and beta4 proteins (*Brenner et al., J Biol Chem., 2000:275:6453-6461*). From the superposition of beta1 and beta4 structures in new **SFig.2B** is also apparent that the largest root-mean-square deviation is in the loop area. This is expected, as beta4 loop is 12 amino acids longer than loop of beta1 (*Brenner et al., J Biol Chem., 2000:275:6453-6461*). Despite this fact, the positioning of the major secondary structure elements in beta1 and beta4 loops is similar. Therefore, we are convinced that Alpha Fold modeling of beta1 protein offers a reasonable degree of reliability whereas beta4 modeling is fully aligned with experimentally obtained structural coordinates. We now include Alpha Fold structure validation into the re-revised manuscript (new **Fig.S2** and p. 21, second Parag., l. 3-6).

6-“*Fig 4B: In the cryo-EM structure the extracellular domains of the neighboring beta4 subunits form an interface. The authors may want to indicate whether and how the docking of PROG is affected by the interface*”.

The reviewer raises here a very interesting point. Indeed, we pointed PROG docking location to the inter-subunit interface between two neighboring beta4 proteins within a BK heterotetramer. Based on this, and on the cryo-EM structure of BK heterotetramers containing beta4 subunits (*Tao and MacKinnon, 2019*), we have added a new figure (**SFig.8**) to this re-revised manuscript to show the expected docking area for a single molecule of PROG as an example at the interface between two adjacent beta4 subunits: while progesterone binds beta-sheets on one beta4 subunit, the steroid faces disordered areas of the neighboring beta4 loop. These disordered areas are expected to have a high degree of spatial flexibility, thus likely favoring the transduction of the conformational changes triggered by progesterone-beta1 binding into modification of BK channel gating and activity. We now include this line of thought in the main text (p. 17, l. 6-15 from bottom of page).

7-“*Fig 4A: in the low affinity site W32F seemed not to disrupt PROG binding, but in the high affinity site W87F acted differently. The authors may want to perform the docking simulations of both W37F and W87F to match with the experimental observations. This would be a demonstration of the validity of the docking approach.*”

As requested, we performed molecular dynamic simulations: representative end points of the trajectories are now presented as new panels **C** and **D** of **SFig.3**.

REVIEWERS' COMMENTS

Reviewer #1 (Remarks to the Author):

The authors added key new data that now clearly shows the physiological significance of their findings. The new version of the manuscript is excellent.

Reviewer #2 (Remarks to the Author):

The authors' responded to all previous questions. However, the answers to some of the questions do not resolve the concerns, as noted in the following.

1. About the role of lipids in the docking results. The author stated difficulties make sense but the concern remains. In order to solve this issue the authors may perform additional studies, such as molecular dynamic simulations of the docking results in the presence of lipids to test the stability of the bound structure.
3. About validation of Prometheus records. SFig 1A, is there any significant difference between beta1+PROG and beta1 boiled?
4. About mutational results on PROG binding sites. The speculations cannot satisfactorily explain the apparently conflicting results.
5. About beta subunits structures used in docking. Would the difference between beta4 structures of experiment and alpha fold at the loop region affect the docking of PROG in that region?
6. About the predicted PROG binding in the subunits interface. Given the interference of the neighboring beta subunits would the docking results be still the same?

Reply to Reviewers

We deeply thank the reviewers for their time and most valuable suggestions when reviewing our re-revised manuscript. Reviewer 1 found this version of our manuscript to be “excellent” and did not raise any additional concern. Reviewer 2 listed several points that in his/her eyes needed to be addressed further. Our point-by-point reply to Reviewer 2 is provided below. We also attached a “red copy” version of the manuscript to facilitate further review by the reviewers and Editors.

Reply to Reviewer 2.

1. *About the role of lipids in the docking results. The author stated difficulties make sense but the concern remains. In order to solve this issue the authors may perform additional studies, such as molecular dynamic simulations of the docking results in the presence of lipids to test the stability of the bound structure.*

We fully agree with the Reviewer that membrane lipids may affect stability of the progesterone-beta1 subunit complexes. The suggestive computational studies will be massive and a study on its own as native BK channels localize both inside and outside lipid rafts of different and highly complex lipid composition (Lam et al., 2004, PMID: 36988883; Weaver et al., 2007, PMID: 17711864; Tajima et al., 2011, PMID: 21135099; Vaithianathan et al., 2023, PMID: 36988883). In addition, the Reviewer does not provide any hint on which membrane lipid species s/he would like to be tested. Moreover, teasing out the influence of membrane lipid composition on “*the stability of the bound structure*” will require simulations with various lipid species that will need to be titrated in order to prove quantitative parameters from concentration-response relationships, as well as establishing lipid structure-response relationships that would provide mechanistic insights on a possible modulatory role of membrane lipids. Therefore, taking into account this very valid concern, and following the Editors’ advice, we now explicitly indicate in the text the need for studies on membrane lipid contributions to progesterone-sensing by BK channels as much needed work for the future (p. 19; l. 2,3).

2. *About validation of Prometheus records. SFig 1A, is there any significant difference between beta1+PROG and beta1 boiled?*

We only noticed clustering of deltaT data-points in “boiled beta1” group but there was no statistically significant difference in deltaT from this group when compared to beta1 that was not subjected to heating-cooling cycle (boiling). This is not surprising as low molecular weight proteins may re-fold in chaperon-free environment (To et al., 2023, PMID: 36417429). Thus, we have added a clarification to the text underscoring that heating-cooling cycling (including boiling) is expected to enhance conformational variability (p. 6; l. 9-11). We have also been trying different approaches (e.g., heat shocks at different temperatures and of variant duration) to maintain protein in unfolded state, and recently determined that this could be achieved by addition of detergent (i.e., SDS titration) to the protein-containing sample immediately before loading of samples in the microscale thermophoresis machine. As detergent ablates protein folds, thermal unfolding plot lacks conformation transitions and data plots look flat, similar to those shown in the “no protein” panel on Supplementary Figure 1c.

3. *About mutational results on PROG binding sites. The speculations cannot satisfactorily explain the apparently conflicting results.*

We recognize that the speculative nature of our explanations may not fully dispel the concern of the reviewer. Thus, taking into account the reviewer's concern and following the Editors' advice, we have added a statement indicating that systematic mutagenesis of beta1 protein is needed in the future to unveil whether amino acids other than those probed in the present study participate in progesterone recognition, and/or in the communication between the two sites initially identified in the current study (p. 10; l. 3,4 from bottom; p. 18, l. 1,2,6,7 from bottom; p. 19, l. 2,3).

4. *About beta subunits structures used in docking. Would the difference between beta4 structures of experiment and alpha fold at the loop region affect the docking of PROG in that region?*

We cannot totally rule out the possibility raised by the reviewer but it is an unlikely scenario. The deviation between AlphaFold-predicted and experimentally obtained structures is about the size of carbon-carbon bond (Supplementary Figure 2a). Pairwise RMSD comparison of two structures results in an averaged deviation of 1.475 Å. In the Supplementary Figure 2 legend we now provide this RMSD measure for the beta4 subunit structures under comparison. It is extremely unlikely that this small difference will significantly impact ligand docking mode(s).

5. *About the predicted PROG binding in the subunits interface. Given the interference of the neighboring beta subunits would the docking results be still the same?*

This is a challenging and exciting question. As our paper mostly focuses on beta1 subunits, we first must point out that beta1 subunit presence in the channel native environment (where putative neighboring beta subunits would be present) is very transient and dynamic. While channel-forming alpha subunits spend hours on the plasma membrane, beta1 subunits only reside steadily in the cell membrane for a few minutes due to the fast exchange between their plasma membrane and intracellular pools, as documented by the group of Jon Jaggar (Leo et al., 2014, PMID: 24464482). We confirmed the dynamic nature of beta1 subunit membrane trafficking in a more recent work demonstrating that cholesterol up-regulates the amount of beta1 protein in cerebral artery myocyte plasma membrane within minutes (Bukiya et al., 2021, PMID: 33556372). Given their transient presence in the native channel complex, it remains currently uncertain whether beta1 subunits interfere with each other and, in particular, whether such interference could alter progesterone binding. It is a limitation of our data that we cannot establish with certainty whether, in the presence of multiple beta1 subunits, progesterone binding would be sequential (and allosteric) or concerted. At this point, it seems premature to experimentally address this point because the very stoichiometry of alpha-beta subunits (initially assumed to be 1:1) in native BK channels is far from established: enrichment of BK channels with different beta subunits does modify the phenotype in a manner consistent with the fact that before such enrichment BK channels (including native BK in cerebrovascular smooth muscle) are unlikely alpha-beta octamers (Wang et al., 2002, PMID: 11880485; Kuntamallappanavar et al., 2017, PMID: 28012000). Second, we cannot ignore the more important interactions between beta1 subunits and channel-forming alphas. Although there seems to be a debate on how close to each other these proteins reside within BK channel complexes (Tao and MacKinnon, 2019, PMID: 31815672), some interference is expected because the presence of beta1 subunits in BK channels undisputedly affects BK current characteristics (Orio et al., 2002, PMID:

12136044; *Li and Yan, 2016, PMID: 27238261*) and membrane lipid pharmacology (*Dopico and Bukiya, 2014, PMID: 25202277; Torres et al., 2014; PMID: 25346693; Dopico et al., 2018, PMID: 29748711*). Last but not least, a potential contribution of gamma subunits to the interaction of the different proteins that conform native smooth muscle BK channel complexes cannot be ruled out (*Evanson et al., 2014, PMID: 24906643; Li and Yan, 2016, PMID: 27238261*). Considering all aforementioned points, extensive docking studies are needed to answer the reviewer's question and its extensive ramifications. Moreover, even if all these docking studies are completed, subsequent molecular dynamics simulations will be needed to establish stability of newly predicted complexes of beta subunits with progesterone. These simulations on large protein complexes will require upgrade on our computing capabilities. Lastly, experimental (phenotyping) validation of computational prediction will also be required.

Nevertheless, to begin to address the reviewer's question, we performed docking of progesterone on beta4 subunits associated with BK alpha subunits associated in heteromeric channel complexes as determined from cryo-EM (*Tao and MacKinnon, 2019, PMID: 31815672*). Our findings show that although positioning of progesterone slightly differs from that rendered by docking onto an isolated beta4 monomer, progesterone still docks on beta4 loop area when the complex is considered. We now updated **Supplementary Figure 8** to reflect this docking result. Considering all the information summarized in this paragraph, we have also added a few words about future work on subunit stoichiometry and its impact on progesterone-sensing by BK channel heteromers (p. 17; 2nd Parag., l. 10-12,16,17).